# An inducible amphipathic *α*-helix mediates subcellular targeting and membrane binding of RPE65

Sheetal Uppal, Tingting Liu, Emily Galvan, Fatima Gomez, Tishina Tittley, Eugenia Poliakov, Susan Gentleman, T Michael Redmond

**RPE65 retinol isomerase is an indispensable player in the visual cycle between the vertebrate retina and RPE. Although membrane association is critical for RPE65 function, its mechanism is not clear. Residues 107–125 are believed to interact with membranes but are unresolved in all RPE65 crystal structures, whereas palmitoylation at C112 also plays a role. We report the mechanism of membrane recognition and binding by RPE65. Binding of aa107–125 synthetic peptide with membrane-mimicking micellar surfaces induces transition from unstructured loop to amphipathic *α*-helical (AH) structure but this transition is automatic in the C112-palmitoylated peptide. We demonstrate that the AH significantly affects palmitoylation level, membrane association, and isomerization activity of RPE65. Furthermore, aa107–125 functions as a membrane sensor and the AH as a membrane-targeting motif. Molecular dynamic simulations clearly show AH-membrane insertion, supporting our experimental findings. Collectively, these studies allow us to propose a working model for RPE65-membrane binding, and to provide a novel role for cysteine palmitoylation.**

## Introduction

Many proteins, including eukaryotic, viral, and bacterial proteins, contain amphipathic *α*-helices (AHs) that function not only as membrane anchors, but which also can mediate other roles such as membrane-targeting motifs, membrane curvature sensing, the level of lipid unsaturation, remodeling of membranes, or as shields to protect membranes or lipid droplets (Gallop et al, 2006; Masuda et al, 2006; Drin et al, 2007; Welker et al, 2010; Copic et al, 2018). It is believed that many peripheral membrane proteins (PMPs) are soluble proteins that interact transiently with the membranes to perform critical functions that ultimately control their residence time on the membrane. Despite recent advances in structural determination methodologies, the structures of very few PMPs have been solved and we lack information about the conformational changes associated with membrane binding (Allen et al, 2019). Moreover, a challenging task is the identification of AHs in such proteins, owing to the complex amino acid compositions and structures of AHs, which are not as well defined as those of transmembrane segments in integral membrane proteins (Tsirigos et al, 2018).

RPE65 retinol isomerase plays a crucial role in the visual (retinoid) cycle of the retina (i.e., the regeneration of 11-*cis* retinal [11cRAL] chromophore) by catalyzing the rate-limiting isomerization step (Redmond et al, 1998, 2005; Jin et al, 2005; Moiseyev et al, 2005). RPE65 converts and isomerizes all-*trans* retinyl esters, mainly all-*trans* retinyl palmitate (atRP) into 11-*cis* retinol (11cROL). Because of the highly lipophilic nature of atRP, it is restricted to the lipid bilayer or to RPE retinosomes (Imanishi et al, 2004; Orban et al, 2011). RPE65 must extract its lipophilic substrate atRP from the smooth endoplasmic reticulum (sER) membrane so there is a continuous supply of 11cRAL for normal vision (Nikolaeva et al, 2009). Previous findings clearly demonstrate that RPE65 is a PMP lacking any clear transmembrane motifs (Hamel et al, 1993a, 1993b; Golczak et al, 2010). Structural inspection of RPE65 reveals a hydrophobic tunnel that connects the protein's exterior to the catalytic center. This may provide a direct route for substrate entry and/or product release. The hydrophobic lining of the interior cavity provides an ideal environment for the passage of lipophilic substrate from the membrane into the catalytic center (Kiser et al, 2009). Several models of RPE65-membrane interaction have been proposed (Golczak et al, 2010; Yuan et al, 2010). However, none so far adequately explains the mechanism of membrane binding of RPE65.

Crystallographic studies show that the loop 107–125 is crystallographically poorly resolved or unresolved in available RPE65 structures, leading to the suggestion that it constitutes a disordered region (Kiser et al, 2009). In contrast, however, when the primary structure of RPE65 was first determined, we identified the region of amino acids (aa) 107–125 as a potential amphipathic *α*-helix and suggested a role for it in RPE65 membrane binding (Hamel et al, 1993b). The aa107–125 sequence also contains a PDPCK motif, highly conserved in metazoan orthologs of the carotenoid cleavage dioxygenase (CCD) family of which RPE65 is a member (Poliakov et al, 2017). Moreover, we have shown recently that

Laboratory of Retinal Cell and Molecular Biology, National Eye Institute, National Institutes of Health, Bethesda, MD, USA

Correspondence: redmondd@nei.nih.gov

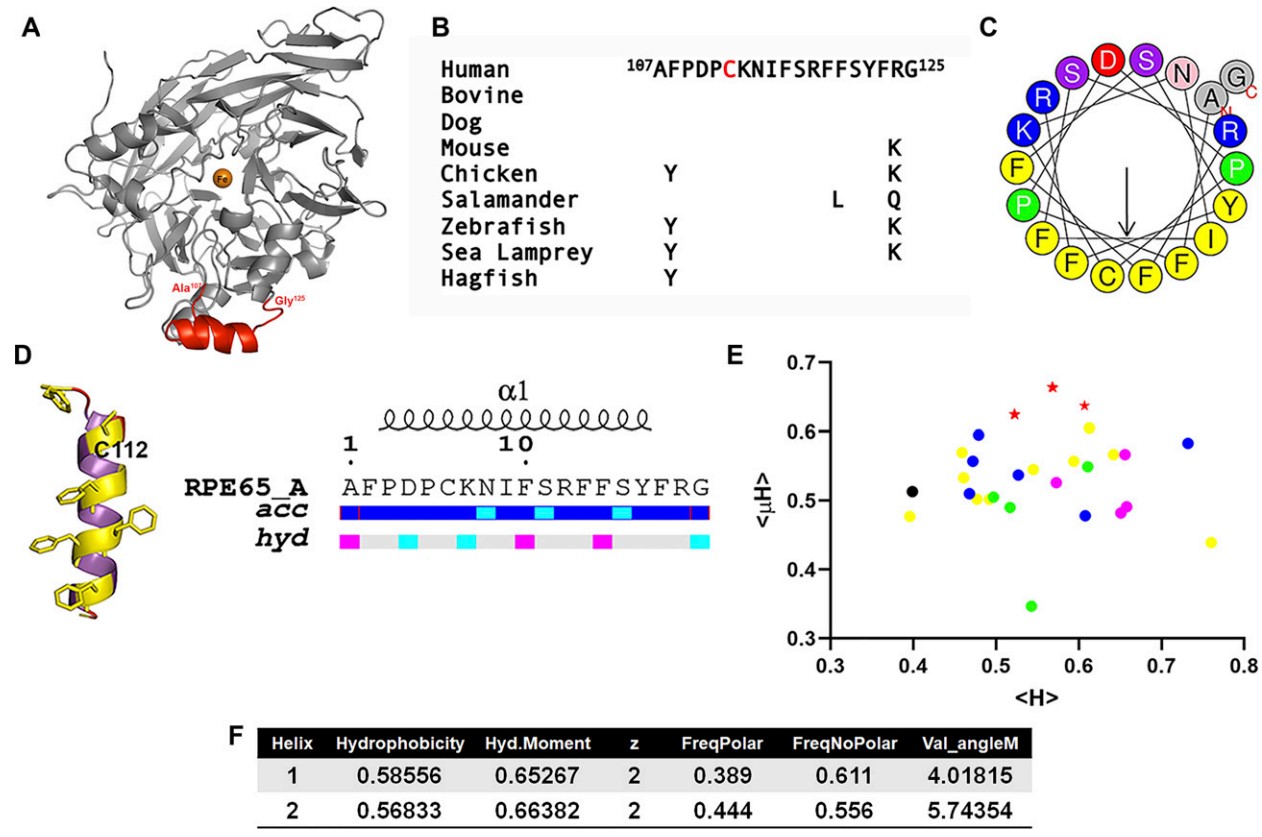

**Figure 1. In silico analysis of the crystallographically unresolved (aa107–125) of RPE65 protein.**
**(A)** Three-dimensional structure of bovine RPE65 (PDB ID: 3FSN). The crystallographically unresolved (aa107–125) region of RPE65 protein, modeled using I-TASSER server, is shown in red color. **(B)** Alignment of RPE65 aa107–125 from several jawed and jawless vertebrate taxa showing sequence conservation throughout evolution. **(C)** Helical wheel representation of the crystallographically unresolved aa107–125 region of RPE65 protein using HELIQUEST server. **(D)** Predicted secondary structure, relative accessibility and hydropathy analysis of the aa107–125 region of RPE65 protein using I-TASSER and ENDScript server, respectively. **(E)** Hydropathic moment plot of combined 35 different RPE65 aa107–125 and paralogous conserved region sequences from vertebrate and non-vertebrate carotenoid cleavage dioxygenase (CCD) paralogs, derived from HeliQuest (Gautier et al, 2008). Red stars, RPE65 s; magenta circles, vertebrate BCO1s; blue circles, vertebrate BCO2s; black circle, early chordate (*Ciona*) CCD; yellow circles, arthropod (*Trinorchestia*, *Palaemon*, *Drosophila*, and *Galleria*) CCDs; and green circles, nematode (*Caenorhabditis*) -CCDs. Note tight grouping of all 10 RPE65 s distinct from other paralogs. **(F)** Physicochemical properties of the crystallographically unresolved (aa107–125) of RPE65 protein. Abbreviations: acc, accessibility; hyd, hydrophobicity.

S-palmitoylation at C112 residue, located in the unresolved region, is important for membrane association and visual cycle function (Uppal et al, 2019a, 2019b).

To resolve this knowledge gap, we hypothesized that the crystallographically unresolved aa107–125 loop of RPE65 acts as a "membrane sensor" that can switch between a disordered loop and a strongly amphipathic helix, and that this AH acts as a membrane targeting motif. To this end, we used chemically synthesized peptides corresponding to the aa107–125 (AH[107–125]) region of RPE65 and studied their structural features and membrane-binding properties using biochemical and biophysical techniques. We generated a set of alanine-substituted mutant proteins to determine the effect of AH[107-125] on the membrane-binding, palmitoylation, and visual cycle function of RPE65. In addition, we examined the ER membrane-targeting feature of the AH[107-125] region using GFP fusion constructs. Last, molecular dynamics simulation studies were carried out to understand the mechanism of binding of AH[107-125] region to lipid membranes. Overall, our work establishes that the AH[107-125] region in RPE65 protein is a key structural element of this protein that serves as an intrinsic membrane-targeting motif and, thus, is essential to the function of RPE65.

# Results

## The crystallographically unresolved region (aa107–125) of RPE65 is intrinsically disordered in aqueous solution

In total, 18 structures of RPE65 from bovine RPE have been solved and deposited in the Protein Data Bank (Kiser et al, 2009, 2012, 2015, 2017; Golczak et al, 2010; Zhang et al, 2015; Blum et al, 2021). Structural analysis of the available RPE65 structures in Protein Data Bank revealed that each structure has a missing region that includes amino acid residues (aa) 107–125 (Fig 1A and Table S1). The aa107–125 region is highly conserved showing minimal conservative changes from jawless to jawed vertebrates down to mammals (Fig 1B). To characterize the structural details of this conserved but crystallographically unresolved region, we performed in silico

studies using HeliQuest and I-TASSER servers (Gautier et al, 2008; Zhang, 2008; Roy et al, 2010; Yang et al, 2015). Helical wheel plotting and in silico secondary structural analysis revealed that this region can form a putative amphipathic alpha (α)-helix with a significant contrast between the hydrophobic and hydrophilic faces (Fig 1C and D). The hydrophilic face contains several basic amino acid residues, whereas the C112 residue is located in the hydrophobic face surrounded by hydrophobic phenylalanine (Phe) residues. A hydropathic moment plot (Eisenberg et al, 1982) shows this sequence to be highly amphipathic and membrane surface-seeking, even compared with other CCDs showing similarity in their regions paralogous to aa107–125 (Fig 1E). Thus, this crystallographically unresolved region of RPE65 is predicted to be surface-accessible and to have a strong amphipathic character with a high hydrophobic moment (Fig 1D–F).

We further biochemically characterized the secondary structure of a chemically synthesized peptide corresponding to the amphipathic helix (RPE65$^{107-125}$) using circular dichroism (CD) spectroscopy. The CD spectrum of AH$^{107-125}$ peptide showed that in aqueous buffer (10 mM sodium phosphate buffer, pH 7.0) and in water at pH 4.5 indicates mainly random coil conformation with a small percentage of secondary structure elements (Fig 2A, left graph and Fig 2D). However, addition of trifluoroethanol (TFE; an α-helical structure stabilizer) to 50% to the RPE65$^{107-125}$ peptide resulted in change from an unstructured to an α-helical structure for the RPE65$^{107-125}$ peptide with a maximum at about 195 nm and two minima at 208 and 222 nm (Fig 2A; [Sonnichsen et al, 1992]). We observed a significant increase in the α-helical content, up to ~95% of the RPE65$^{107-125}$ peptide in the presence of 50% TFE compared with aqueous buffer and water at pH 4.5 (Fig 2D). These results allowed us to corroborate the secondary structure prediction analyses and to conclude that the crystallographically unresolved region at aa107–125 of RPE65 has a propensity to form an α-helical structure. Since we reported earlier that palmitoylation at C112 residue is involved in membrane association, we also synthesized and characterized a palmitoylated form of the RPE65$^{107-125}$ peptide in which a palmitoyl moiety is attached to C112 via a thioester linkage (Uppal et al, 2019a). To our surprise, the palmitoylated RPE65$^{107-125}$ peptide exhibited an α-helical structure even in aqueous buffer and in water at pH 4.5 (Fig 2A, right graph). Addition of TFE to 50% resulted in a change in the shape of the CD spectrum and an increase in the amplitude, indicating further stabilization of the α-helical structure of the C112-palmitoylated AH peptide (Fig 2A, right graph and Fig 2D).

### Unstructured to amphipathic α-helical structural transition of crystallographically unresolved region of RPE65 in presence of membrane mimetics

Next, we examined the secondary structure of the synthetic RPE65$^{107-125}$ peptide in detergents which can mimic the membrane environment. We found that the CD spectra of the non-palmitoylated RPE65$^{107-125}$ peptide exhibits an α-helical conformation (~95%) in the presence of 20 mM SDS or 20 mM $n$-dodecyl $\beta$-D-maltoside (DDM) (Fig 2B, left graph and Fig 2D). Structural transition of the non-palmitoylated RPE65$^{107-125}$ peptide from random coil to α-helical conformation upon interaction with

detergent micelles clearly indicate that the RPE65$^{107-125}$ peptide has a binding affinity for lipids. However, in the case of the palmitoylated RPE65$^{107-125}$ peptide, we observed no change in the shape of the CD spectra and no substantial increase in the α-helical content in presence of detergent (Fig 2B, right graph and Fig 2D), corroborating the "locked" α-helical nature of the C112-palmitoylated AH peptide.

We next analyzed the secondary structure of the synthetic RPE65$^{107-125}$ peptides in the presence of liposomes. Similar to the previous results with detergent micelles, CD spectroscopy revealed that the non-palmitoylated RPE65$^{107-125}$ peptide adopts an α-helical conformation in the presence of dioleoylphosphatidylcholine (DOPC) liposomes and an increase in helicity was observed with an increasing molar ratio of liposomes (Fig 2C, left graph).

Gel filtration chromatography was used to further investigate the interaction of the RPE65$^{107-125}$ peptide with detergent. We found that non-palmitoylated RPE65$^{107-125}$ peptide dissolved in DDM micelles elutes at ~17.9 min, which corresponds to a molecular weight of ~49 kD that is roughly equal to the molecular weight of detergent micelles. In contrast, in the absence of detergent, non-palmitoylated RPE65$^{107-125}$ peptide elutes at ~30 min which is much later than expected, considering its low molecular weight (2,300 D; Fig 2E, left panel). This suggests that the non-palmitoylated peptide interacts with the chromatographic media, probably via hydrophobic interactions. Indeed, detergent was necessary to elute completely all the non-specifically bound non-palmitoylated peptide from the gel filtration column. When CD spectra of the eluted peptide were recorded, a significant change from unstructured to α-helical structure of the non-palmitoylated peptide was detected in the presence of DDM detergent micelles (Fig 2E, insets in left panels). In contrast, the gel filtration profiles of C112-palmitoylated RPE65$^{107-125}$ peptide were quite different. In the presence of DDM, the C112-palmitoylated peptide elutes earlier (~17.6 min) compared with the peptide in the absence of detergent (~23.1 min; Fig 2E, right panels and insets). However, the latter elution time is much earlier than that of the non-palmitoylated peptide in the absence of detergent (Fig 2E, left panel). These data indicate further stabilization of the α-helical structure of the palmitoylated peptide upon binding to the detergent micelles. The calculated molecular weights of the detergent micelle-bound non-palmitoylated and palmitoylated RPE65$^{107-125}$ peptide from the calibration curve is ~50 kD (Fig S1). These data are concordant with a strong propensity of this peptide to bind lipid-mimicking detergents.

### Binding of AH$^{107-125}$ peptides and purified recombinant maltose-binding protein (MBP)-AH$^{107-125}$ with liposomal membranes

To extend these studies we next performed a sucrose gradient liposome co-flotation assay. In this assay, membrane-binding proteins float during centrifugation with liposomes in the upper, lower density fraction, whereas soluble proteins remain in the lower, higher density fraction (Fig 3A). A fusion construct of 6xHis-MBP and RPE65$^{107-125}$ AH peptide (6xHis-MBP-AH$^{107-125}$) was generated and over-produced heterologously in *Escherichia coli*. The 6xHis-MBP-AH$^{107-125}$ protein was purified to homogeneity with an apparent molecular weight of ~46 kD on SDS–PAGE gel, slightly higher than the purified 6xHis-MBP protein (Fig 3B). For the assay, 6xHis-MBP

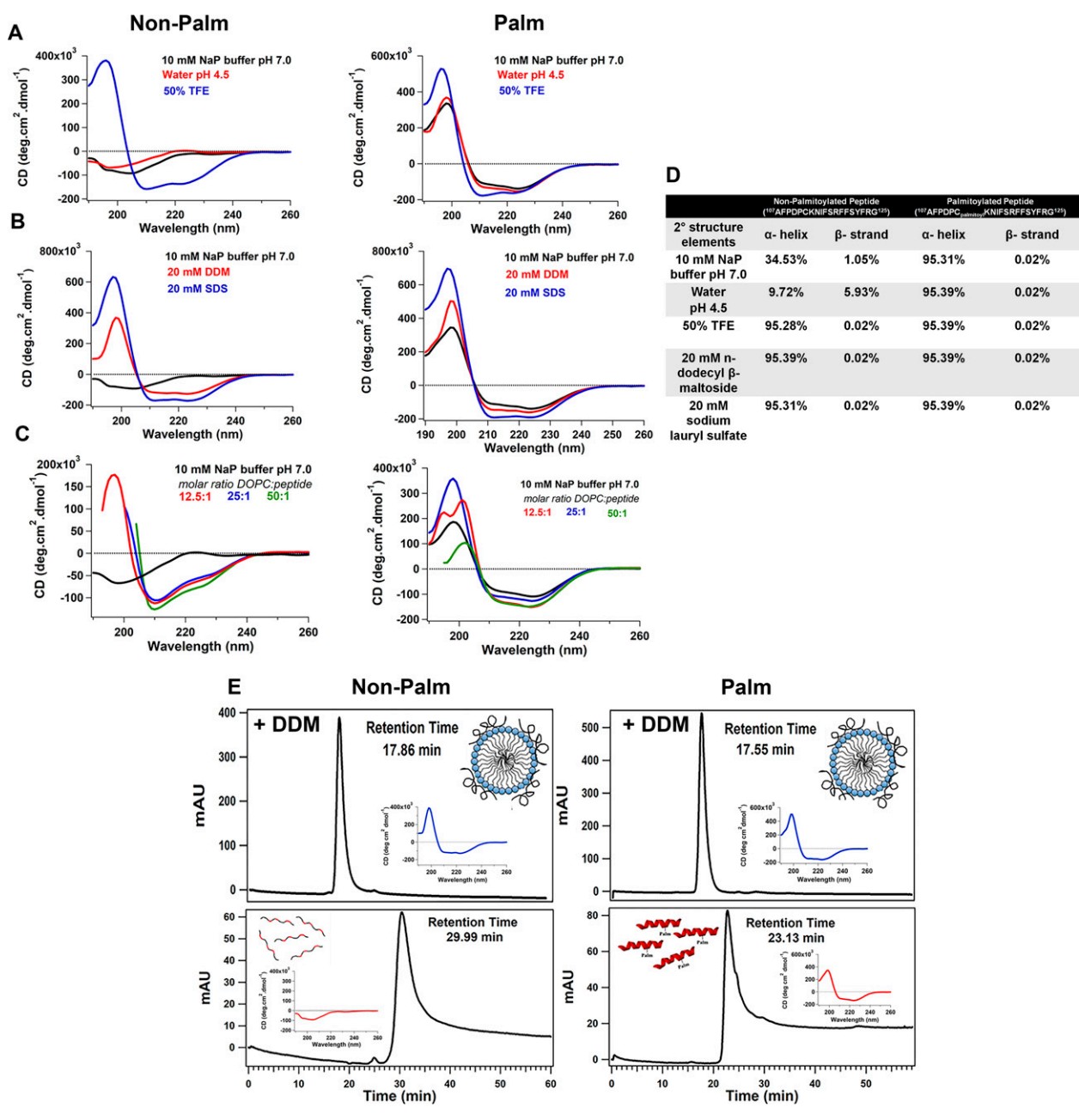

**Figure 2. The crystallographically unresolved aa107–125 region forms an amphipathic α-helical structure upon association with membrane.**
**(A)** Far-UV CD spectra of chemically synthesized aa107–125 non-palmitoylated (left) and palmitoylated (right) peptides were recorded at a peptide concentration of 100 μM. **(A, B, C)** Secondary structure analyses were carried out in (A) 10 mM sodium phosphate buffer pH 7.0, water pH 4.5, and 50:50 TFE/water solution, (B) 10 mM sodium phosphate buffer, pH 7.0, supplemented with either 20 mM n-dodecyl-β-D-maltoside (DDM) or 20 mM SDS, (C) in presence of DOPC liposomes. **(D)** Percentage composition of secondary structural elements of chemically synthesized non-palmitoylated and palmitoylated peptide under different buffer conditions. **(E)** Analytical gel filtration of non-palmitoylated (left panels) and palmitoylated (right panels) RPE65[107-125] peptides pre-incubated with 10 mM detergent for 30 min at 4°C. Detergent-incubated peptides with were chromatographed on a Protein PAK 200SW column equilibrated with 10 mM sodium phosphate buffer, pH 7.4, containing the detergent in which the peptides were incubated. Insets show CD spectra recorded for the eluted peptides. Abbreviations: NaP, sodium phosphate; TFE, trifluoroethanol; NaP, sodium phosphate.

and 6xHis-MBP-AH[107-125] purified proteins were mixed, separately, with fluorescent 1-Oleoyl-2-[12-[(7-nitro-2-1,3-benzoxadiazol-4-yl)amino]dodecanoyl]-sn-Glycero-3-Phosphocholine (NBD:PC-DOPC) liposomes and resolved in sucrose density gradients. After centrifugation, the upper fractions containing the liposomes and the middle and bottom fractions were collected and subjected to

SDS–PAGE and Coomassie staining. As shown in Fig 3C and D, 6xHis-MBP-AH[107-125] protein showed an increased level of protein (~5-fold) compared with the low level of control 6xHis-MBP protein in the liposome-containing upper fraction, indicating the binding of RPE65[107-125] peptide to the liposomal membranes. Furthermore, we assessed the presence of liposomes in each fraction and observed

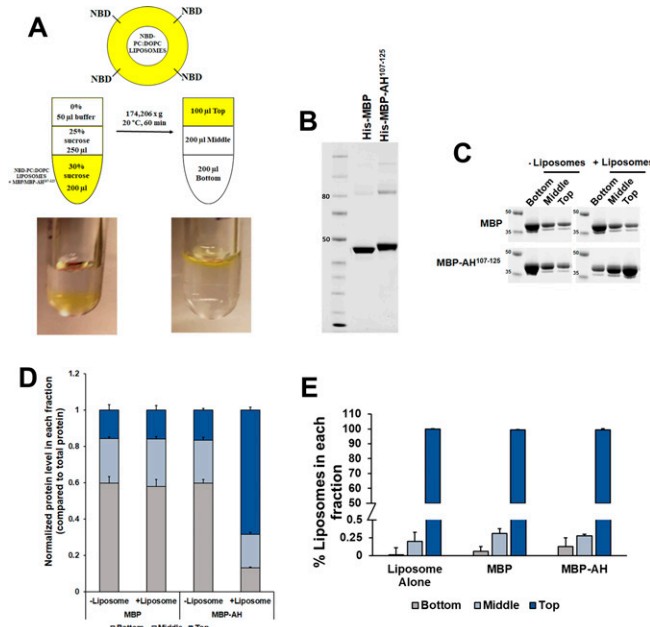

**Figure 3. Recombinant maltose-binding protein (MBP)-AH[107-125] fusion protein interacts with liposomes.**
**(A)** Schematic representation of the liposome flotation assay used for determining interactions between MBP protein or MBP-AH[107-125] fusion construct and liposomes. **(B)** SDS–PAGE of purified MBP and of a MBP-AH[107-125] fusion protein. The purification was carried out using affinity and gel filtration chromatography and analyzed by electrophoresis on 4–12% Bis–Tris SDS-polyacrylamide gel. **(C)** All three fractions collected from liposome flotation experiments were analyzed for protein by SDS–PAGE and Coomassie staining. The experiments were performed in triplicate. **(D, E)** Bar graphs shows the protein quantification (D) and percentage of liposomes (E) in each fraction ±SD. Abbreviations: AH, amphipathic helix; DOPC, 1,2-dioleoyl-sn-glycero-3-phosphatidylcholine; MBP, maltose-binding protein; NBD/NBD:PC, 1-Oleoyl-2-[12-[(7-nitro-2-1,3-benzoxadiazol-4-yl)amino]dodecanoyl]-sn-glycero-3-phosphocholine.
Source data are available for this figure.

that almost ~100% of liposomes were fractionated in the top fraction upon centrifugation (Fig 3E).

### The amphipathic aa107–125 region of RPE65 is an ER membrane-targeting motif

It is well known that RPE65 protein is primarily localized to RPE smooth ER (microsomes) (Hamel et al, 1993a), so we generated GFP-AH[107-125], a GFP fusion construct of the RPE65[107-125] peptide to assess whether this amphipathic region is sufficient to target a heterologous protein to the ER membrane. The GFP-AH[107-125] and GFP only constructs were transfected into COS-7 cells, and we analyzed the GFP subcellular distribution using confocal laser scanning microscopy. As expected, GFP showed a strong nuclear and diffused cytoplasmic distribution. However, GFP-AH[107-125], although reduced in the central nucleus, showed a staining pattern that included the nuclear membrane, but was strongest in the perinuclear region, and extended in a reticular pattern throughout the cytoplasm corresponding primarily to the ER membrane (Fig 4A).

We further confirmed the ER membrane distribution of the GFP-AH[107-125] construct by immuno-colocalization experiments. As shown in Fig 4B, the GFP-AH[107-125] construct showed an ER membrane fluorescence pattern which colocalized with the ER membrane marker DsRed2-ER. Nonetheless, the fluorescence intensity of GFP was increased roughly fourfold in the ER compared with nucleus of cells expressing the GFP-AH[107-125] construct (Fig 4C). In addition, native RPE65 protein displayed an ER colocalization pattern coincident with DsRed2-ER. Based on the above findings, we conclude that the RPE65[107-125] amphipathic helix serves as an intrinsic determinant for subcellular targeting and membrane association.

### The AH[107-125] hydrophobic face is essential for the membrane binding and isomerase activity of RPE65 protein

Having established that RPE65 uses an amphipathic helix for membrane targeting, we performed a systematic mutagenesis of the AH[107-125] region to determine the role of the amphipathic helix in the membrane association and visual cycle function of RPE65. First, we investigated how individual amino acid residues in the AH[107-125] region affect protein expression compared with wild-type RPE65 protein. We found that the expression level of AH[107-125] RPE65 mutant proteins were mostly comparable with the wild-type protein except for the P111A and R118A mutant proteins (Fig 5A).

Next, we assessed the membrane association of the RPE65 protein in the AH[107-125] mutant proteins. Subcellular fractionation studies revealed that almost all the AH[107-125] mutant proteins showed a decrease in membrane association compared with the wild-type protein (Fig 5A). In the case of the F108A, D110A, P111A, and R118A mutant proteins, we observed a loss of RPE65 protein in the membrane fraction, indicating that mutation of these residues in the AH significantly affect the RPE65-membrane association. Next, we investigated the effect of alanine substitution of the amino acids in AH[107-125] on RPE65 palmitoylation. We subjected the membrane fraction of the HEK293-F cells overexpressing mutant proteins to the acyl-resin assisted capture (acyl-RAC) assay to determine palmitoylation. We observed a significant reduction in the palmitoylation level of the AH[107-125] mutants with very faint or no protein band detected in the hydroxylamine (HAM)-treated sample compared with wild-type RPE65 protein, which showed a prominent protein band in the HAM-treated sample (Fig 5B).

Although these results clearly showed that mutations in the AH[107-125] region perturb the membrane association and palmitoylation level of RPE65, we considered the possibility that they also would affect the enzymatic activity of RPE65. To test this, we performed isomerase activity assays (i.e., the formation of 11-*cis* retinol) for AH[107-125] mutant proteins using a heterologous non-RPE cell culture system that mimics the visual cycle. Our results revealed that substitution of any of almost all the hydrophobic amino acid residues in the AH[107-125] region of RPE65 has a severe effect (>50% reduction) on isomerase activity compared to wild-type protein (Fig 5C). In addition to hydrophobic residues, we found that some of the polar/charged residues of the AH (D110, N114, R118, and R124) are important to RPE65 enzyme activity. Moreover, we observed that mutation of residue C112 completely abolished the membrane association of RPE65, thus affecting the palmitoylation and isomerization activity of RPE65, supporting our previous results (Uppal et al, 2019a). Taken together, our data strongly suggests that

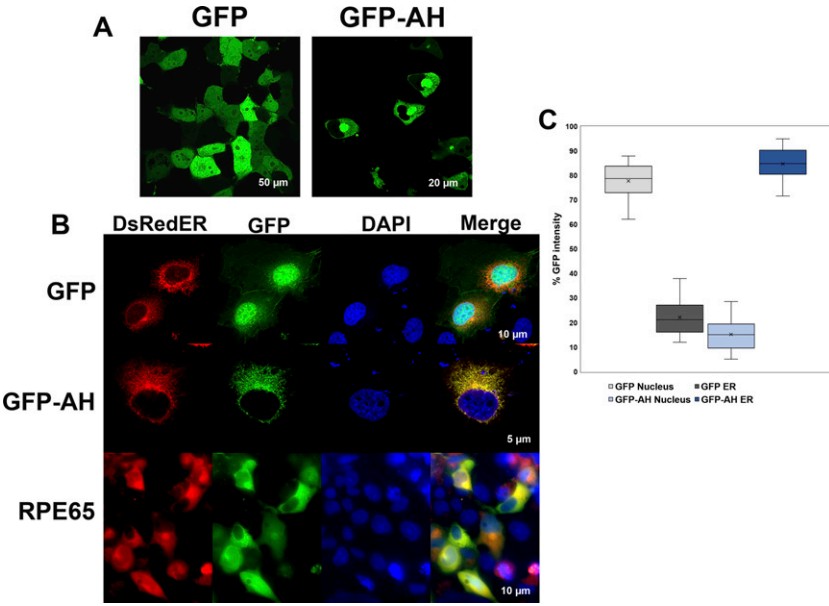

**Figure 4.  RPE65 AH[107-125] region is sufficient for targeting to endoplasmic reticulum (ER) membrane.**
**(A)** COS-7 cells transiently transfected with plasmids containing GFP or a GFP-AH[107-125] fusion construct were analyzed using live-cell imaging for localization of GFP. GFP showed a strong nuclear and diffused cytoplasmic distribution (left), whereas localization of GFP-AH[107-125] was largely reduced in the nucleus but with strong perinuclear region and ER membrane localization (right). **(B)** COS-7 cells grown on coverslips were co-transfected with plasmids containing GFP only or GFP-AH[107-125] fusion construct and DsRedER, an ER localization marker. At 48 h post-transfection, cells were paraformaldehyde-fixed on coverslips and analyzed by confocal laser scanning microscopy. In addition, COS-7 cells grown on coverslips were co-transfected with pVitro2-dRPE65/CRALBP and DsRedER and cultured for 48 h and processed for immunolabeling with monoclonal mouse antisera against RPE65 protein. Bound primary antibodies were probed with fluorescently labeled donkey-anti-mouse 488 secondary antibody and slides were analyzed by confocal laser scanning microscopy. The GFP-AH[107-125] construct, unlike GFP alone, colocalized with the ER membrane marker DsRed2-ER, in a similar pattern to that of native RPE65 (see merged lanes). **(C)** The background corrected percentage of GFP fluorescence intensity corresponding to nucleus and ER from GFP- and GFP-AH[107-125]–expressing cells are plotted in a box and whisker diagram. Boxes represent the lower and upper quartiles, the marker in the box (x) indicates the median of each series, and the whiskers represent minimum and maximum values.

the AH is a critical structural element in the membrane binding of RPE65 and it can be assumed that the hydrophobic face of the AH interacts with the lipid bilayer.

### Patient-identified mutation affects the RPE65 protein expression and isomerase activity

The *RPE65* gene is subject to many mutations which give rise to the early onset severe blinding disorder Leber congenital Amaurosis type 2 (LCA2). In this regard, several pathogenic mutations (missense, indels, or duplications) or variants of unknown significance (VUS; listed in Table S2) centered in the amphipathic region of RPE65 have been identified in humans (Fokkema et al, 2011). 7 of 19 residues of AH[107-125] are involved in missense mutations, almost 40% of the residues. We believe that these mutations/VUSs would be likely to affect the membrane association and thus, enzymatic function of RPE65. To confirm this, we generated site-directed mutants of RPE65 where we substituted these residues with their specific altered residue and expressed these in HEK293-F cells. Before analyzing the membrane association of these suspect mutants, we determined the levels of protein expression in comparison with the wild-type RPE65 protein. Surprisingly, almost all the pathogenic mutations except N114H and I115T significantly lower the protein expression level (Fig S2).

### Molecular dynamics simulation of amphipathic helix of RPE65 in a membrane environment

Our biochemical data clearly show that the amphipathic α-helix, irrespective of the palmitoyl moiety, can interact with the lipid membrane. To further investigate the membrane-binding property of the AH[107-125] region, we performed molecular dynamics (MD) simulations in a membrane-mimetic environment (see the Materials and Methods section). The peptide was initially positioned away from the lipid bilayer (Fig 6 and *SV1*) and the system was simulated for 1 μs (Video 1). During the initial phase of the simulation, the peptide remained diffused within the solvent but quickly encountered the membrane surface within the first 20 ns, with the Z-distance between the COM of the peptide and bilayer dropping from ~60 to ~30 Å (Fig 6). The peptide remained bound at the surface of the membrane for the next ~350 ns (with 28.1 Å average COM Z-distance between the peptide and membrane) forming interactions with the lipid headgroups (Fig 6). This was followed by a deeper transition (~around 370–450 ns) of the peptide into the lipid bilayer with COM Z-distance between the peptide and membrane going below 20 Å. Thereafter, the peptide remained stably inserted within the membrane for the rest of the simulation (with 12.6 Å average COM Z-distance between the peptide and membrane) with the hydrophobic face of the amphipathic helix pointed towards the lipid tails, and the polar residues interacting with the lipid headgroups (Fig 6).

## Discussion

Interaction with RPE smooth ER membrane plays a crucial role in the function of RPE65 in the visual cycle; however, the structural basis of its membrane binding remains unclear (Nikolaeva et al, 2009; Kiser, 2021). Unfortunately, none of the available crystal structures of RPE65 has provided any functional insight into how RPE65 interacts with membranes to fulfil its functional role. Here, we describe the membrane-sensing mechanism whereby the crystallographically unresolved region of RPE65 accomplishes this role. We demonstrate the key importance of this region in RPE65 protein membrane binding showing that it is an intrinsically

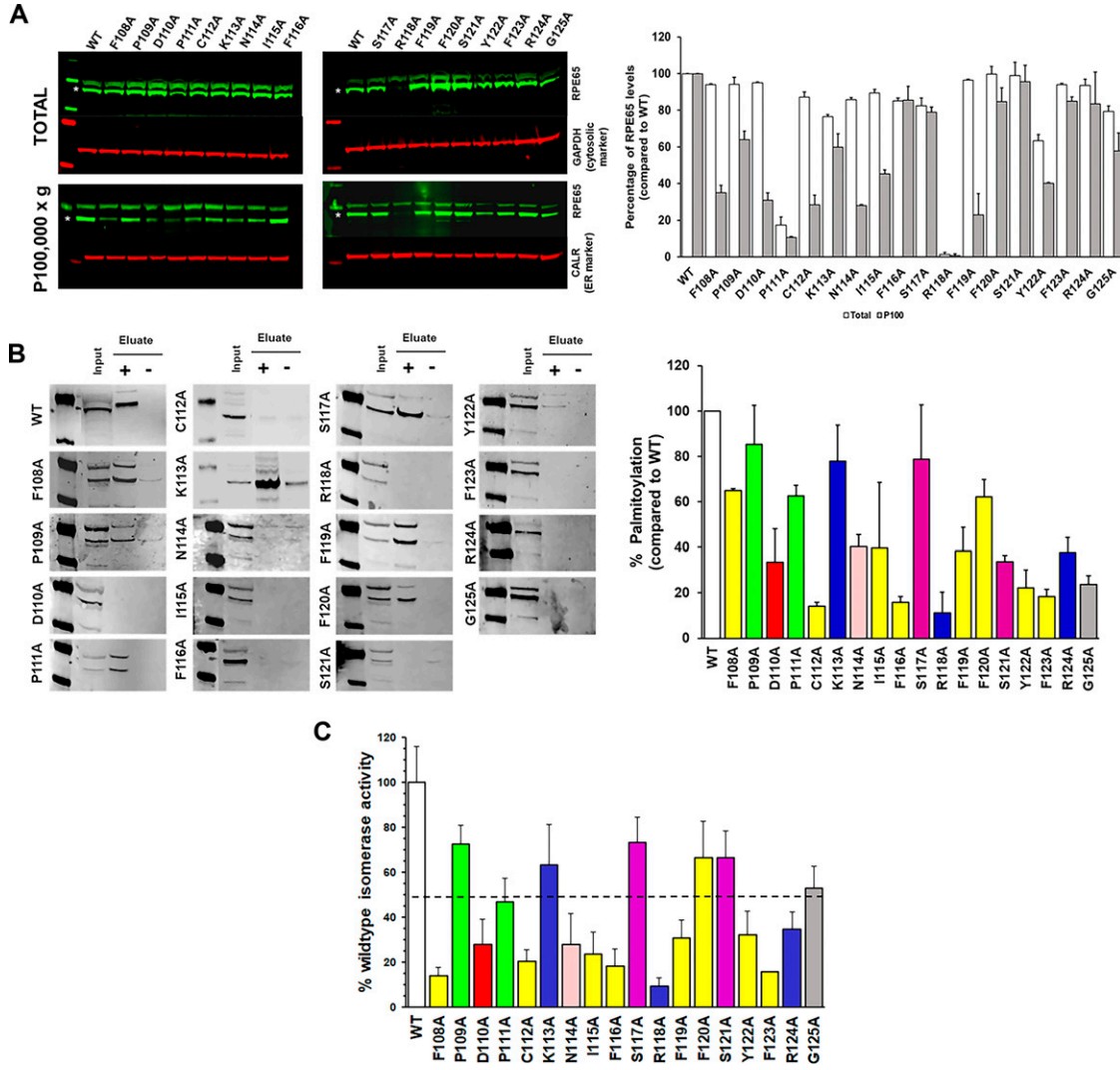

**Figure 5. AH$^{107-125}$ region influences palmitoylation, membrane binding, and isomerization activity of RPE65 protein.**
**(A)** HEK293F cells overexpressing wild-type RPE65 and RPE65 proteins harboring alanine mutants of each of the AH$^{107-125}$ region residues were analyzed for total protein level and membrane-associated level (P 100,000$g$) using sucrose subcellular fractionation, followed by SDS–PAGE and Western blotting with primary antibody to RPE65. GAPDH and calreticulin were used as loading controls for cytosolic and membrane fractions, respectively. Data represented as mean ± SD from three independent experiments. **(B)** HEK293F cells overexpressing wild-type RPE65 and RPE65 proteins harboring alanine mutants of each of the AH$^{107-125}$ region residues were analyzed for palmitoylation level using the acyl-resin assisted capture assay followed by immunoblotting for RPE65 using input and pulldown eluted samples. Samples were treated with 0.5 M hydroxylamine (HAM; indicated as "+") or 0.5 M NaCl (indicated as "–"), respectively. Results were calculated as mean ± SD from three independent experiments. The vertical line in some of the blots indicates a splice. **(C)** In vitro isomerase activity was assessed for HEK293F cells overexpressing wild-type RPE65 and RPE65 proteins harboring alanine mutants of each of the AH$^{107-125}$ region residues. Data were represented as mean ± SD from three independent experiments. Dashed line represents 50% isomerase activity. Source data are available for this figure.

disordered region (IDR) that can fold into an amphipathic α-helix upon its interaction with membranes. The aa107–125 IDR/AH structural transition we describe is key to RPE65 function. Also, it is not irrelevant that the aa107–125 sequence and its AH is highly conserved (Fig 1) back to the origins of RPE65 in jawless vertebrates (Poliakov et al, 2012), underscoring its importance to RPE65 function. Whereas the AH appears to be the primary means by which RPE65 as a PMP binds to RPE smooth ER, other surface residues or patches of residues may play a role (Kiser & Palczewski, 2010). Also, RPE65 missense mutations distant from aa107–125 have the capacity to modify its folding/stability in ways that may disrupt its membrane binding (Wu et al, 2022).

Previous attempts by other groups to explain how RPE65 binds to RPE smooth ER membranes to accomplish its function lacked this information on the pivotal role of this "missing" aa107–125 region and its structural transition. Although Xue et al (2004) proposed a "palmitoylation switch" mechanism to account for RPE65's dual nature of soluble ("sRPE65") and membrane-associated forms ("mRPE65"; [Xue et al, 2004]), it was on the basis of data later shown to be erroneous (Takahashi et al, 2009). Despite this false lead, palmitoylation at C112 was shown to play an important role in RPE65 membrane association and enzymatic function (Uppal et al, 2019a). Golczak et al (2010) in a study on the importance of membrane structure on RPE65 enzymatic activity, presented a crystallographic

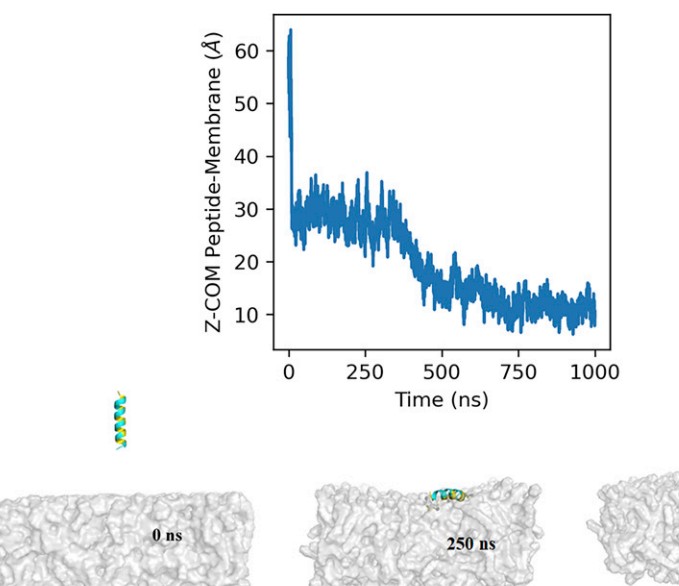

**Figure 6.  Molecular dynamics simulation of RPE65 AH[107–125] interaction in membrane environment.**
The graph shows the Z-component of the distance between the center of mass of the AH[107–125] peptide and the lipid bilayer with respect to time. The insets represent snapshots of the interaction of the AH[107–125] helix region of RPE65 with the lipid bilayer at different time points (0, 250, and 1,000 ns during the 1-μs simulation time).

model of RPE65. Residues aa109–126, which could not be experimentally modeled in their crystal structure, were depicted as an a-helix embedded in the membrane. However, no experimental evidence was offered to support this conjecture. Finally, Yuan et al (2010), while presenting evidence that the major mechanism for RPE65-membrane binding was due to electrostatic interactions between patches of basic residues on the surface of RPE65 and acidic phospholipid headgroups, further postulated that the "disordered hydrophobic loop" sequence—FFSYF—(aa119–123), contained within aa107–125, could act as an anchor to stabilize RPE65's interaction with membranes. This 5-residue sequence would be too short to accomplish this task.

It has been reported in several PMPs that IDRs can mediate highly specific and reversible binding to cell membranes (Cornish et al, 2020). Structurally, IDRs lack well-defined 2° or 3° structures and have been shown to undergo structural transition from disordered-to-ordered state upon binding with a protein partner or lipid membrane (Clore, 2014; Granata et al, 2015; Zea et al, 2016), often in a previously unanticipated way. For example, in the phosphatidylinositol phosphate kinase family, a previously known structure has been shown to subserve an additional purpose: the known activation loop of phosphatidylinositol phosphate kinases also acts as an amphipathic membrane-sensor in phosphatidylinositol 4-phosphate 5-kinase (PIP5K; [Liu et al, 2016]). The amphipathic α-helix is well suited for membrane-induced conformational change because of the segregation of the polar and non-polar residues between the two faces of the helix (Gimenez-Andres et al, 2018). An amphipathic helix would allow the transition of PMPs like RPE65 protein between the membrane bound and unbound state, consistent with previous biochemical observations which showed that a fraction of total RPE65 protein is soluble (Hamel et al, 1993a). This duality of the aa107–125 IDR/AH structure also explains its absence in RPE65 crystal structures where its flexibility would contribute to large apparent B-factors (Kiser et al, 2009; Golczak et al, 2010).

The universe of the AH repertoire is wide—it is known that some AHs favor binding to membranes that are curved (Bigay et al, 2005; Drin et al, 2007), have packing defects and/or increased surface charge (Cornell, 2016), or to lipid droplets (Pataki et al, 2018; Olarte et al, 2022), among others. Also, a cholesterol sensing AH in squalene monooxygenase has been identified that converts into a degron when ER membrane cholesterol becomes high (Chua et al, 2017). ER membranes are a part of a cellular lipid "territory" comprising ER–nuclear envelope (NE)-cis-Golgi, whereas the other major "territory" centers on trans-Golgi–plasma membrane (PM) endosomes (Bigay & Antonny, 2012; Jackson et al, 2016). The difference between the two relates to lipid packing—the former being characterized by membranes with little anionic charge and loose packing of "conical" and "cylindrical" phospholipids and low sterols and sphingolipids (i.e., with "packing defects"), whereas the latter have high levels of "cylindrical" phospholipids, sterols, and sphingolipids, and so are quite tightly packed (Jackson et al, 2016). It is also known that different kinds of AH localize to different membrane regions, depending both on the relative charge of the polar face residues and on the relative size and hydrophobicity of the hydrophobic face residues (Pranke et al, 2011; Bigay & Antonny, 2012). Does the particular conformation of the RPE65 AH[107–125] target it to a particular region of the RPE ER? We do not have the answer to this question, but experiments to address this are underway. However, we can conclude that the AH[107–125] sequence targets to the ER rather than to the PM territory.

Importantly, we show that the aa107–125 AH alone can translocate GFP protein to the ER compartment (Fig 4), diverting it from the nucleus, implying that the AH is necessary and sufficient to target the RPE65 protein to the ER membrane and, thus, that it behaves as an intrinsic membrane-targeting motif. The hydrophobic side chains and some of the charged basic residues in the AH play a critical role in the RPE65-membrane binding and isomerization activity. Our molecular dynamics simulations (Fig 6) provide detailed molecular insight into the binding and insertion of

the AH peptide into the membrane. We see that the initial interaction of AH peptide with the membrane is mediated by interactions of charged basic residues with the polar headgroups of the lipids. Later in the process, the side chains of hydrophobic residues interact with the hydrophobic core of the membrane allowing the insertion of the AH peptide into the lipid bilayer. In addition, our MD simulation results are strongly supported by previous data showing the affinity of RPE65 protein for acidic phospholipids via electrostatic interactions (Yuan et al, 2010). In addition, the RPE65 AH[107-125] has a high hydrophobic moment, which is also a characteristic feature of surface-active antimicrobial peptides, which further indicates that the AH[107-125] region is capable of inserting into the lipid bilayer (Li et al, 2017). Amphipathic helices of such surface-active antimicrobial peptides are capable of significantly perturbing membrane structure. Membrane perturbation, therefore, may be a means by which RPE65 can more efficiently access its substrate atRP in the sER membrane. This is important because the actual orientation/location of atRP in bilayers is not known. Various carotenoids have been studied in this regard, with hydrocarbon carotenoids (such as β-carotene) residing in the central hydrophobic core and hydroxycarotenoids (such as lutein) oriented more transversely with their hydroxylated rings among the polar phospholipid heads and their polyene chains in the interior (Woodall et al, 1997; Jemiola-Rzeminska et al, 2005). It is possible that atRP might orient with its more polar ester linkage towards the aqueous surface with its hydrocarbon retinyl and lipid chains towards the interior. Alternatively, the entire molecule may be located in the central cleavage plane of the membrane. In this case, deep, but reversible, embedding of RPE65 in the membrane may be required for it to access its substrate.

The uniquely important role of palmitoylation at C112 in this process must be addressed. We show that whereas the synthetic unpalmitoylated peptide acquires α-helical structure in the presence of membrane mimetics, the palmitoylated peptide exhibits α-helical structure even in water or buffer (Fig 2). Thus, palmitoylation "locks" the sequence into an a-helical conformation such that the hydrophobic residues face all in one direction, which in turn increases membrane residence time. This effect of direct local conformational switch to an AH is previously unreported for any protein, expanding the repertoire of roles of cysteine palmitoylation. The approximate converse, where an AH directs an adjacent cysteine palmitoylation, has been found to occur in the electrogenic sodium/calcium exchanger 1 (NCX1; [Plain et al, 2017]), but this is distal rather than local in effect. This finding further emphasizes the crucial role of C112 in RPE65 structure and function (Takahashi et al, 2009; Uppal et al, 2019a).

Based on our findings, we propose a hypothetical working model of RPE65-membrane binding as illustrated in Fig 7. Most of the residues, including C112, in the RPE65 AH[107-125] are highly conserved, underscoring its importance. In the membrane-unbound state, the unstructured region (RPE65[107-125]) targets the soluble inactive RPE65 protein to the ER membrane, which is the functional localization compartment for RPE65. The association of the RPE65 IDR with the membrane is primarily mediated by electrostatic interactions through the positively charged amino acids which results in the conformational change of the IDR to an amphipathic α-helix. The 2° structure transition in the IDR to an AH is followed by interaction of the hydrophobic residues of the AH to the hydrophobic

lipid core of the membrane and thus, insertion of the AH deep into the membrane bilayer. We hypothesize that this insertion of the AH along with the palmitoylation of C112 residue is the critical step in the proper orientation of substrate-binding cleft onto the membrane allowing RPE65 to extract its highly lipophilic substrate, atRP. This might facilitate binding to the membrane and in recognition of the substrate for uptake into the substrate-binding cleft. Upon isomerization of the substrate to 11-*cis* retinol, RPE65 protein can undergo self- or enzyme-mediated deacylation followed by conformational change to facilitate product release and, ultimately, release of RPE65 from the membrane. How does this proceed and what is the functional role of desorption? We know from our previous work (Uppal et al, 2019a, 2019b) that RPE65 palmitoylation is reduced in the presence of active LRAT. In that article, we speculated that the palmitoyl group could be donated to/removed by LRAT to esterify all-*trans* retinol. Alternatively, acyl-protein thioesterase–mediated deacylation may occur. Either case would weaken the AH and, therefore, RPE65 could disassociate from the membrane. We speculate that this desorption could be required for transfer of 11-*cis* retinol from the binding tunnel of RPE65 to 11-*cis*–specific cellular retinal-binding protein (CRALBP; RLBP1). CRALBP is an obligate component of the visual cycle, playing a key role in the downstream processing of 11-*cis* retinol (Saari et al, 1994). The 11-cis retinol is thought to exit the active site first (Kiser et al, 2015). How 11-*cis* retinol transfers from RPE65 is currently unclear: whether it is passed directly for oxidation to RDH5 in the membrane phase or if it is passed to CRALBP in the soluble phase, which then presents it to RDH5 for oxidation. The latter scenario is more likely as absence of CRALBP stalls progress of isomerization (Saari et al, 2001). Upon transfer of its 11-*cis* retinol product to CRALBP, RPE65 would be ready for another isomerization cycle. These possibilities require clarification.

In conclusion, this study provides important new details on how RPE65 binds to sER membranes in its visual cycle function as retinol isomerase. We reveal the key role of the crystallographically unresolved aa107–125 region as an intrinsic membrane-sensing and targeting motif. Furthermore, palmitoylation of Cys112 (in the center of the hydrophobic face) also, uniquely, expedites AH formation, suggesting an additional role of cysteine palmitoylation in protein targeting. With this information, we are now seeking to identify the cognate palmitoyltransferase enzyme that might be involved in the palmitoylation of C112 in the AH region of RPE65 and the fate of the palmitoyl moiety after the release of the RPE65 from the membrane. Furthermore, MD simulations studies of RPE65 protein with membrane containing its substrate, atRP, will shed light on the molecular details of the RPE65-membrane interaction and how substrate is extracted from within the lipid bilayer for catalysis. These experiments are currently ongoing.

# Materials and Methods

### Sequence analysis and modeling of the unresolved aa107–125 region of RPE65

The physicochemical properties and secondary structure of the crystallographically unresolved region (aa107–125) of RPE65 were

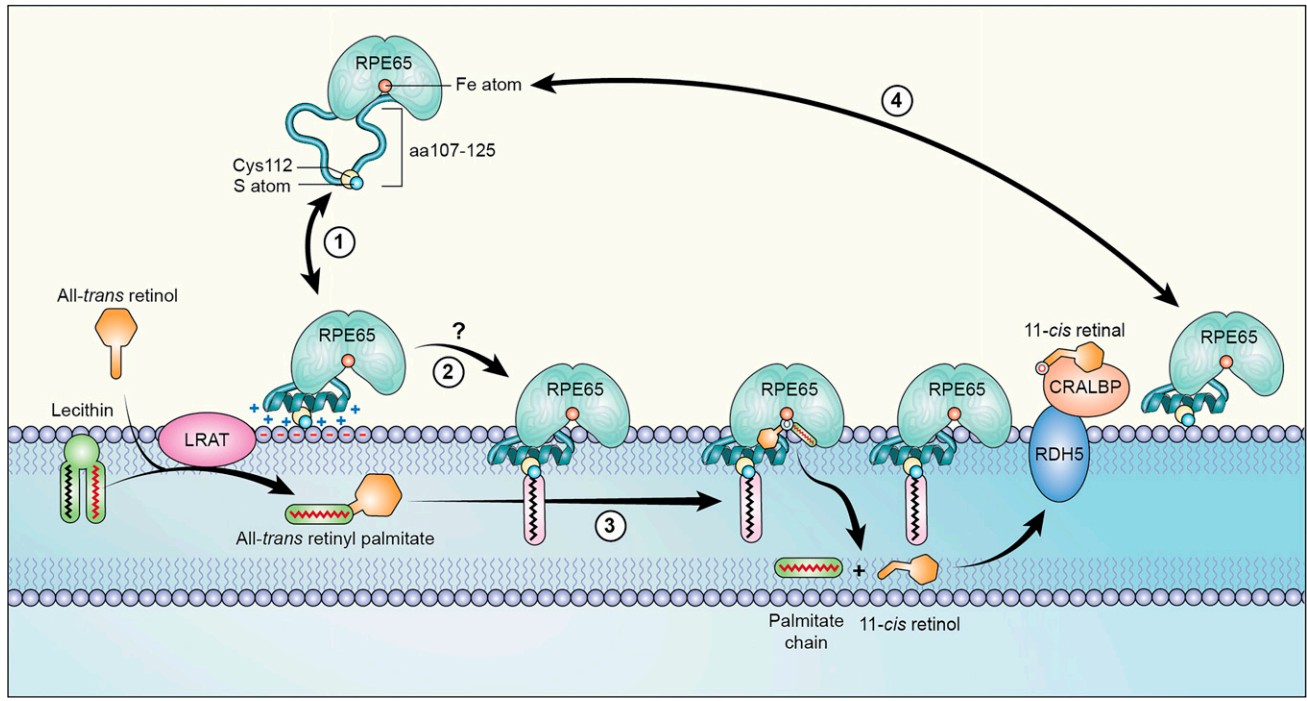

**Figure 7. Cartoon model of RPE65-ER membrane binding.**
We propose that the aa107–125 sequence is the key structural element of RPE65 that serves as an intrinsic membrane-targeting motif thereby regulating the function of RPE65 in the visual cycle: **(1)** As an intrinsically disordered region, the aa107–125 loop targets soluble-phase RPE65 to the ER membrane and the association with membrane results in a structural change from intrinsically disordered region to an amphipathic alpha-helix (AH); **(2)** the initial association is mediated by electrostatic interactions of the positively charged AH with the acidic headgroups of membrane phospholipids, followed by insertion of the AH deep into the lipid bilayer in concert with palmitoylation of the C112 residue located in the AH region (putatively by an unknown DHHC palmitoyltransferase), thereby converting RPE65 to its membrane-associated form; **(3)** RPE65 now has access to and can take up its substrate all-*trans* retinyl palmitate (at-RP; generated by the enzyme lecithin:retinol acyltransferase) isomerizing it to 11-*cis* retinol. 11-*cis* retinol is bound by CRALBP and presented to RDH5 for ultimate oxidation to the chromophore 11-*cis* retinal; **(4)** RPE65 may remain membrane associated or be released following depalmitoylation.

modeled using HeliQuest (https://heliquest.ipmc.cnrs.fr/) and I-TASSER (https://zhanggroup.org/I-TASSER/) online servers, respectively (Gautier et al, 2008; Roy et al, 2010; Yang et al, 2015; Yang & Zhang, 2015). The secondary structure of the crystallographically unresolved region was visualized using PyMOL visualization software (Schrödinger, Inc.). The secondary structure, relative accessibility, and hydropathy of crystallographically unresolved region (aa107–125) of RPE65 was generated using web servers ENDscript (https://endscript.ibcp.fr) (Robert & Gouet, 2014).

## Peptide synthesis

Peptides corresponding to the crystallographically unresolved region were synthesized in two forms: non-palmitoylated ([107]AFPDPCKNIFSRFFSYFRG[125]; molecular weight 2,300.66 g/mol) and palmitoylated ([107]AFPDPC[112]palmitoylKNIFSRFFSYFRG[125]; molecular weight 2,539.74 g/mol) using the solid-phase method using Fmoc (*N*-9-fluorenylmethoxy-carbonyl) chemistry and were purified by reverse-phase HPLC (Bio-Synthesis Inc.). The purity of each peptide was >95%, as determined by mass spectroscopy and reverse-phase HPLC. The peptides were supplied as lyophilized powder and were stored at –80°C for long-term use.

## Preparation of synthetic liposomes

1,2-Dioleoyl-sn-glycero-3-phosphatidylcholine (*DOPC*; Avanti Polar Lipids) was dissolved in chloroform to prepare a 25 mg/ml stock (~32 mM). The desired lipid amount (1.5 mg/ml, ~2 mM) was transferred to a 15 ml Pyrex glass tube and dried under argon gas using a TurboVap evaporation system (Caliper). Lipid films were resuspended in pre-warmed volumes of 1× PBS buffer and rehydrated by incubation at 37°C for 2 h. Rehydrated liposomes were subjected to five freeze–thaw cycles using liquid nitrogen/warm water (40°C) followed by extrusion using a mini-extruder device (Avanti Polar Lipids) with 20 passages through polycarbonate membrane (pore size 0.1 μm, diam. 19 mm; Whatman Nucleopore Track-Etched membranes). The extruded liposomes (~2 mM) were collected and transferred into sealed fresh glass tubes for further use.

## Phospholipid estimation

The total phospholipid content of the liposomes was measured colorimetrically using an EnzyChrom phospholipid assay kit (Bio-Assay Systems). The liposomal samples were diluted in 0.5% Triton X-100 (Sigma Millipore).

## CD spectroscopy

The lyophilized synthetic peptides were dissolved in organic solvent (DMSO). to 15 mg/ml (~6 mM). The accurate concentration of peptides was confirmed using the molar extinction coefficient calculated by the Expasy ProtParam (https://web.expasy.org/protparam/) tool (Gasteiger et al, 2005). The extinction coefficient of the peptide at 280 nm is 1,490 $M^{-1}cm^{-1}$. The desired working concentrations were achieved by diluting the stock solutions with 10 mM sodium phosphate buffer, pH 7.0, 20 mM SDS, 20 mM n-dodecyl β-D-maltoside (DDM; Sigma Millipore), 50% tri-fluoroethanol (TFE; Sigma Millipore), or in DOPC liposomes. In the case of DOPC liposomes, different liposomes concentration (1.25, 2.5, and 5 mM) were mixed with 100 μM peptides in a molar ratio of 12.5:1, 25:1, and 50:1, respectively. Changes in the secondary structure of the synthetic peptides in these different solvents were analyzed by CD spectroscopy on a Chirascan Q100 spectropolarimeter (Applied Photophysics) using a cylindrical quartz cell of path-length 1 mm at 260–190 nm (far-UV region) with a 1-nm increment, at 25°C. Three consecutive spectral scans were collected, averaged, and corrected by subtracting the corresponding blank spectrum without the peptide, and subjected to noise reduction. The spectral units were converted from millidegree (mdeg) to molar residue ellipticity (MRE, deg $cm^2$ $dmol^{-1}$) using peptide molar concentrations. The secondary structural elements were calculated using the K2D3 online server (Kelly et al, 2005).

## Gel filtration chromatography

Analytical gel filtration chromatography was performed on Protein-PAK 200SW column with dimensions of 8.0 mm (i.d.) × 30 cm (glass; P/N WAT011786; Waters Associates) equilibrated with chromatographic buffer containing 10 mM sodium phosphate buffer, pH 7.4. 100 μl of peptide solution corresponding to 0.1 mM concentration was loaded into the equilibrated column and run at a flow rate of 0.5 ml/min on an ĀKTA purifier system (Cytiva). In the presence of the detergent micelles, the column was equilibrated with the chromatographic buffer containing 10 mM n-dodecyl β-D-maltoside (CMC in $H_2O$: ~0.17 mM) and the peptides were incubated in the corresponding detergent for 30 min at 4°C prior loading onto the column. Gel filtration protein standards (Bio-Rad) were used for calibration of the column equilibrated with chromatographic buffer.

## Construction of MBP-AH[107-125] for expression in *E. coli*

We cloned the crystallographically unresolved region (107–125 aa, AH[107-125]) of RPE65 protein into the MBP fusion cloning vector pET His6 MBP TEV LIC (2M-T; a gift from Scott Gradia, plasmid # 29708; Addgene). The forward and reverse primers used for amplification of the RPE65 AH[107-125] region contain ligation-independent cloning v1 tags at the 5′ end (Table S3). The final plasmid construct encoding 6xHis-MBP-AH[107-125] protein contains a His-tagged MBP plus a TEV digestion site (ENLYFQ↓S) fused to the N terminus of the AH[107-125] region. The sequence of the construct after cloning was validated by DNA sequencing (Psomagen).

## Expression and purification of recombinant MBP and MBP-AH[107–125] protein

*E. coli* BL21 (λDE3) plysS cells (Invitrogen) were transformed with the plasmid vector encoding 6xHis-MBP-AH[107-125] protein. For expression, transformed competent cells were grown in terrific broth media containing 100 μg/ml ampicillin and 34 μg/ml chloramphenicol at 37°C at 200 rpm. Once $O.D._{600nm}$ reached 0.8, protein expression was induced using 1 mM iso-propyl β-D-thiogalactopyranoside (IPTG; Gold Bio) and expression was continued for a further 14 h at 37°C. Harvested cell pellets were washed with ice-cold protein purification buffer (50 mM sodium phosphate containing 100 mM NaCl and 1 mM EDTA, pH 7.4) and stored at –20°C. Cells were thawed on ice and resuspended in ice-cold lysis buffer (50 mM sodium phosphate, 100 mM NaCl, and 1 mM EDTA, pH 7.4 containing 100 μg/ml lysozyme, 10 μg/ml DNase, 10 μg/ml PMSF, and 1X protease inhibitor cocktail). Cell lysate was incubated at 25°C for 30 min with end-over-end mixing followed by sonication using SONICS Vibra-Cell (Probe: Model CV334; 10% amplitude; 10 s pulse and 30 s rest for 5 min). Cell nuclear debris was removed by centrifugation at 6,000*g* for 30 min at 4°C. The supernatant containing the soluble 6x-His-tagged MBP-AH[107-125] protein was subjected to affinity chromatography using HiTrap TALON crude, followed by size exclusion chromatography using HiLoad 16/60 Superdex 200 on an ĀKTA purifier system. The 6xHis-MBP protein was expressed and purified using the same method as described above and used as a control protein. The purity of the MBP and MBP-AH[107-125] proteins were assessed by SDS–PAGE and stained with Blazin' Blue protein gel stain (Gold Bio).

## Liposome flotation assay

Binding of the AH[107-125] region to liposomes was examined by a flotation assay. DOPC liposomes were fluorescently labeled with NBD:PC (1-Oleoyl-2-[12-[(7-nitro-2-1,3-benzoxadiazol-4-yl)amino]dodecanoyl]-sn-Glycero-3-Phosphocholine; Avanti Polar Lipids, Cat. no. 810133) at a ratio of 95:5 in chloroform, respectively. We followed the same procedure as outlined above to prepare the NBD:PC-DOPC liposomes. Throughout the procedure the tubes were covered with aluminum foil to protect the fluorescent liposomes from light. For the experiment, 1 mM NBD:PC-DOPC liposomes and 50 μM purified 6XHis-MBP or 6XHis-MBP-AH[107-125] proteins were mixed in 100 μl in 50 mM sodium phosphate buffer, pH 7.4, and incubated for 1 h at room temperature. After incubation, 100 μl of 60% sucrose dissolved in 1× PBS buffer was added and gently mixed with the sample to yield a final concentration of 30% sucrose. Next, 250 μl of 25% sucrose dissolved in 1× PBS buffer was carefully layered on top of the 30% sucrose fraction, and subsequently 50 μl 1× PBS buffer was layered on top of the 25% sucrose fraction, yielding 500 μl of total volume. Sucrose density gradient centrifugation was conducted for 1 h at 20°C at a speed of 174,206*g* in a TL-100 ultracentrifuge using a TLA 120.1 rotor (Beckman Coulter). The top (100 μl) containing the fluorescent liposomes, middle (200 μl), and bottom (200 μl) fractions were collected using microlance 3 steel needles (BD) and mixed with 5× LDS buffer and boiled for 5 min at 95°C, and then 10 μl of the sample was subjected to SDS–PAGE using gradient gels (4–12% Bis-Tris NuPAGE; Invitrogen). Presence of the protein in all the three fractions was visualized by staining gels with Blazin' Blue protein gel stain (Gold Bio). We measured the

phospholipid content of the liposomes in all three collected fractions using EnzyChrom phospholipid assay kit (BioAssay Systems).

## Site-directed mutagenesis

The plasmid pVitro2 encoding the dog *RPE65* cDNA was used as template to generate single mutants of RPE65 protein where each amino acid residue from region 107–125 was substituted to an alanine residue using Quick-Change Lightning site-directed mutagenesis kit (Agilent). The mutagenesis primers listed in Table S3 were designed as per the manufacturer's instructions. The mutations were confirmed by DNA sequencing (Psomagen) and the plasmids were purified by QIAGEN plasmid purification kits (QIAGEN).

## Cell culture and transfection

Cell culture methods and transient transfection protocols were used as described previously (Redmond et al, 2005). Briefly, HEK293-F FreeStyle (293-F; Invitrogen) were grown in serum-free FreeStyle 293 expression medium (Invitrogen) with shaking at 130 rpm in a 37°C incubator under 8% $CO_2$. The HEK293-F cells were transfected using 2 ml of OptiMem-I reduced serum medium containing 40 $\mu$l of 293Fectin reagent (Invitrogen) with 20 $\mu$g of each expression plasmid. Cells were harvested after 48 h transfection.

## Subcellular protein fractionation

Harvested 293-F cells overexpressing wild-type and mutant RPE65 proteins were washed with 1× PBS buffer and resuspended in 0.33 M sucrose in 10 mM phosphate buffer, pH 7.4, containing complete protease inhibitor cocktail. The resuspended cells were disrupted using $N_2$ cavitation under high pressure (700 pounds per square inch [psi]; Parr Instrument Co.) and the lysates were centrifuged at 300g for 10 min at 4°C to remove cell debris and nuclei. The post-nuclear supernatant was then centrifuged at 30,000g for 20 min at 4°C and the resultant supernatant was centrifuged at 100,000g for 1 h at 4°C. The resulting microsomal pellet was resuspended in 0.3% CHAPS in phosphate buffer (10 mM, pH 7.4) containing complete EDTA-free protease inhibitor cocktail and subjected to immuno-blotting using antibodies to RPE65 (1:2,000; custom made [Redmond & Hamel, 2000]) GAPDH (cytosolic marker; GTX100118; GeneTex) and calreticulin (ER marker; Abcam).

## Palmitoylation detection by acyl-RAC assay

The palmitoylation status of wild-type and mutant RPE65 proteins was detected using the (acyl-RAC) method as described previously with minor modifications (Forrester et al, 2011). Briefly, 293-F cells overexpressing the wild-type and $AH^{107-125}$ mutant RPE65 proteins were washed twice with ice-cold 1× PBS buffer, resuspended in buffer A (50 mM Hepes pH 7.4, 150 mM NaCl, 5 mM EDTA, and 1× complete protease inhibitor cocktail [Roche Diagnostics]) and subjected to lysis using $N_2$ cavitation under high pressure (700 pounds psi). The cell lysates were subjected to centrifugation at 900g for 10 min, 4°C and the collected supernatant was further

centrifuged at 20,000g for 30 min, 4°C. The membrane pellet containing RPE65 protein was dissolved in buffer A containing 0.5% Triton X-100 detergent, and 500 $\mu$g of the membrane fraction was used for the assay. In the assay, free cysteine sulfhydryl groups of the protein were blocked using 1% (vol/vol) S-methyl methanethiosulfonate (MMTS; Sigma-Aldrich) prepared in 2× blocking buffer (0.1 M Hepes, 1 mM EDTA, and 2.5% [vol/vol] SDS) and incubated for 15 min at 40°C with frequent vortexing. Subsequently, the proteins were precipitated by acetone precipitation and the pellet resuspended in 0.1 M Hepes, 5 mM EDTA, and 1% SDS (vol/vol) buffer, pH 7.4. A fraction of the solubilized pellet was saved as the input and the remaining fraction was divided into two equal halves: one half was treated with 0.5 M hydroxylamine (HAM; Sigma-Aldrich) to cleave thioesters, and the resulting free cysteine residues captured by activated 4-thiol sepharose beads (Sigma-Aldrich). The other half of the fraction was treated with 0.5 M NaCl and used as an untreated control. Both samples were incubated for 2 h at room temperature with end-over-end mixing. After incubation, the samples were centrifuged at 3,000g for 1 min, the supernatant was removed and retained as the "unbound" fraction. The beads containing the bound protein were washed five times with 1 ml buffer A. Subsequently, the "bound" proteins were eluted from the beads by incubating with 50 $\mu$l Laemmli sample buffer (LSB:BME: Buffer A 0.9:0.1:3) for 15 min at room temperature and then heating at 95°C for 5 min. The input and bound samples from control untreated and HAM-treated were subjected to SDS–PAGE, followed by Western blotting. The experiments were repeated at least three times.

## Protein determination and immunoblot analysis

Total protein content in cell lysate was estimated by Pierce Coomassie Protein Plus assay reagent (Thermo Fisher Scientific) with bovine serum albumin (Sigma-Aldrich) as a standard. Samples were combined with 4 × LDS sample buffer, denatured samples were separated by SDS–PAGE using 4–12% gradient BisTris NuPAGE (Invitrogen) and electro-transferred to nitrocellulose membrane using an iBlot2 gel transfer device (Thermo Fisher Scientific). Membranes were blocked with Intercept blocking buffer (LI-COR Biosciences) for fluorescent Western blotting and then probed with primary antibodies in Intercept blocking buffer for overnight at 4°C. The membranes were washed three times for 5 min each with 1 × TBS containing 0.1% Tween 20, incubated with IRDye 680RD and 800CW secondary antibodies (LI-COR Biosciences, NE; 1: 15,000) in Intercept blocking buffer for 1 h at room temperature and then washed three times with 1 × TBS containing 0.1% Tween 20. For detection, membranes were scanned on an Odyssey CLx Infrared Imager (LI-COR Biosciences) in the 700- and 800-nm channels. The primary antibody used was rabbit polyclonal anti-RPE65 antibody (1:2,000; custom made) (Redmond & Hamel, 2000).

## Construction of GFP-AH$^{107-125}$ for expression and imaging in mammalian cells

Using the pSELECT NGFP-zeo (InvivoGen) expression vector we expressed the 19 amino acid residues of the crystallographically unresolved region ($AH^{107-125}$) fused in frame to the N-terminal of

GFP protein. The expression construct was verified by DNA sequencing (Psomagen). Live cell imaging was performed using COS-7 cells grown on six-well glass bottom black-framed plates (# 1.5 high performance cover glass; 0.17 ± 0.005 mm; Cellvis). Cells were transfected with 2 $\mu$g of GFP or GFP-AH[107-125] plasmid using Fugene 6 transfection reagent (Promega Corporation) in a 1:6 DNA:Fugene 6 ratio and grown at 37°C with 5% $CO_2$ in air. For co-localization studies, COS-7 cells were grown on 1.8 cm poly-lysine coated coverslips (PLL; Neuvitro Corporation) and transfected with GFP or GFP-AH[107-125] plasmid along with DsRedER plasmid used as an ER localization marker. At 48 h post transfection, coverslips with COS-7 cells were washed three times with 10 mM PBS, pH 7.4, and fixed with fixative solution (4% paraformaldehyde in PBS pH 7.4; Electron Microscopy Sciences) for 15 min at room temperature. Coverslips were mounted on microscopic slides using Prolong Glass Antifade with DAPI Mountant (Thermo Fisher Scientific) and stored at 4°C until imaged. In addition, COS-7 cells grown on coverslips were co-transfected with 2 $\mu$g of pVitro2-dRPE65/CRALBP and cultured for 48 h and processed for immunolabeling with monoclonal rabbit RPE65 antibody (1: 250; Abcam). Bound primary antibodies were probed with fluorescently labeled donkey-anti-rabbit 488 secondary antibody (LI-COR Biosciences). We stained the ER compartment using an ER staining kit (Red Fluorescence Cytopainter, Abcam) slides were analyzed by confocal laser scanning microscopy. Imaging was performed using an inverted fluorescence microscope (LSM 700 confocal microscope; Zeiss) equipped with four solid-state lasers (405, 488, 555, and 639 nm). Photomicrographs were taken with a 40× oil immersion lens/1.4-NA using Zeiss ZEN software. Each experiment was performed on a minimum of three biological replicates, with 5–10 fields of view containing on average one to five cells per field of view analyzed. The fluorescence intensity of GFP in the nucleus and ER of 25 cells were analyzed using ImageJ (NIH). We marked the area around the nucleus and the ER region in each cell and obtained the area and the background corrected fluorescence intensity of the marked region. The data were plotted in a box and whisker diagram.

### Isomerization assay

HEK293-F cells heterologously expressing the minimal visual cycle components with RPE65 wild-type and mutant proteins and incubated with atROL were centrifuged and retinoids extracted from the harvested cells under dim red light as previously described (Redmond et al, 2005). Extracted retinoids were saponified and the resultant isomeric retinols were analyzed on 5-$\mu$m particle LiChrospher (Alltech) normal phase columns (2 × 250 mM) on a HPLC system with a UV-visible diode-array detector (Agilent 1,100/1,200 series, Agilent Technologies), following a standard method (Landers & Olson, 1988) as modified by us (Redmond et al, 2005). Data were analyzed using ChemStation32 software (Agilent).

### MD simulation of AH[107-125] peptide in membrane environment

The 19-mer peptide corresponding to the AH[107-125] region (AFPDPCKNIFSRFFSYFRG) of RPE65 protein, modeled as an $\alpha$-helical conformation using the I-TASSER server, was used as input to the CHARMM-GUI membrane builder functionality to setup the system

for MD simulations (Jo et al, 2008). Simulations were performed with a bilayer containing 95% 1,2-Dioleoyl-sn-glycero-3-phosphatidylcholine (DOPC) and 5% 1,2-diheptanoyl-sn-glycero-3-phosphocholine (C7DHPC) further solvated with TIP3P waters, 0.15 M NaCl, and additional neutralizing ions. The peptide was placed at a minimum distance of ~20 Å from the surface of the lipid bilayer, with the final system (with a box dimension of 101 × 101 × 137 Å) consisting of the peptide, 284 DOPC, 16 C7DHPC, 31,038 water molecules, 83 sodium ions, and 85 chloride ions, totaling 134,016 atoms. The simulation was performed using Gromacs 2020.2 and used CHARMM36m force field with a timestep of 2 fs (Brooks et al, 2009; Lee et al, 2016; Huang et al, 2017). The Gromacs input files for equilibration and production provided by CHARMM-GUI were used. The system was first minimized followed by a series of NVT and NPT equilibration steps that consisted of gradual removal of the restraints from lipid and protein atoms as suggested by CHARMM-GUI for a total time of 2 ns. This was followed by a 1-$\mu$s unrestrained NPT production simulation. During the production run, system temperature and pressure were maintained at 300 K and 1 bar, respectively, using the Nose–Hoover thermostat for temperature coupling and Parrinello–Rahman barostat for pressure coupling (Parrinello & Rahman, 1981; Braga & Travis, 2005). Trajectory was visualized using VMD and figures were prepared in Pymol (Humphrey et al, 1996). Gromacs gmx distance command was used to calculate the Z-component of the distance between the center of mass of the peptide and lipid bilayer. *This work was performed using the NIH HPC Biowulf cluster.*

## Supplementary Information

## Acknowledgements

This work was supported by the Intramural Program of the National Institutes of Health, National Eye Institute. We are thankful to Dr. Robert N Fariss (Biological Imaging Core, NEI) for microscopic imaging training and advice. We thank Jacob Nellisserry, Todd Duncan, and Jeanee Bullock for their assistance with the gel filtration chromatographic experiments. We thank Uchechi Nwaneri, NIH Summer Intern, for providing assistance with the patient-identified mutation experiments. We thank Di Wu (Biophysics Core Facility, NHLBI) for assistance with the CD spectroscopy experiments. We thank Dr. Anirban Banerjee, NICHD NIH, for his constructive insights. This study used the high-performance computational capabilities of the Biowulf Linux cluster at the National Institutes of Health (https://hpc.nih.gov).

### Author Contributions

S Uppal: conceptualization, data curation, formal analysis, investigation, methodology, and writing—original draft, review, and editing.
T Liu: data curation, formal analysis, and investigation.
E Galvan: investigation.
F Gomez: investigation.
T Tittley: investigation.

E Poliakov: conceptualization, data curation, formal analysis, methodology, and writing—review and editing.

S Gentleman: conceptualization and writing—review and editing.

TM Redmond: conceptualization, data curation, formal analysis, supervision, funding acquisition, investigation, and writing—review and editing.

## Conflict of Interest Statement

The authors declare that they have no conflict of interest.

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
