## [Reviewer comments · Life Science Alliance]

Life Science Alliance

An Inducible Amphipathic α -Helix Mediates Subcellular Targeting and Membrane Binding of RPE65

Sheetal Uppal, Tingting Liu, Emily Galvan, Fatima Gomez, Tishina Tittley, Eugenia Poliakov, Susan Gentleman, and T. Michael Redmond

DOI: <https://doi.org/10.26508/lsa.202201546>

Corresponding author(s): T. Michael Redmond, National Institutes of Health

Review Timeline:

Submission Date:	2022-06-02
Editorial Decision:	2022-07-11
Revision Received:	2022-09-02
Editorial Decision:	2022-09-22
Revision Received:	2022-09-27
Accepted:	2022-09-28

Scientific Editor: Novella Guidi

Transaction Report:

July 11, 2022

Re: Life Science Alliance manuscript #LSA-2022-01546-T

Dr. T. Michael Redmond
National Eye Institute
6 Center Drive
Building 6, Room 117A
Bethesda, Maryland 20892

Dear Dr. Redmond,

Thank you for submitting your manuscript entitled "An Inducible Amphipathic α -Helix Mediates Subcellular Targeting and Membrane Binding of RPE65" to Life Science Alliance. The manuscript was assessed by expert reviewers, whose comments are appended to this letter. We invite you to submit a revised manuscript addressing the Reviewer comments.

Thank you for this interesting contribution to Life Science Alliance. We are looking forward to receiving your revised manuscript.

Sincerely,

B. MANUSCRIPT ORGANIZATION AND FORMATTING:

Reviewer #1 (Comments to the Authors (Required)):

RPE65 retinol isomerase is essential in the visual cycle of the retinal pigmented epithelium (RPE) in vertebrate retina. Membrane association is critical, although how it is associated to the membrane remains unclear. The authors of this work analyzed the amino acid residues aa107-125 of RPE65, an inherently disordered region of the protein in the crystal structure, to uncover the role of this region in the overall function of the protein. Through elegant and classical biochemical and biophysical techniques, they have found that the aa107-125 can act as a "membrane sensor" that can switch between an amphipathic helix (AH) and a disordered loop. The authors assessed membrane association of the AH region using alanine-scanning mutagenesis of the protein in the AH region, and found that for four mutants (F108A, D110A, P111A, and R118A) they observed a loss of RPE65 protein in the membrane fraction through subcellular fractionation assays, as well as a severe loss in RPE65 function in the presence of those mutants. The circular dichroism (CD) data of the AH with and without the palmitoylation at C112 are compelling and show the propensity of the palmitoylated peptide to display a significantly ordered AH with palmitoylation. Through confocal laser scanning microscopy of transfected tissue culture cells expressing GFP fusions of the AH, the authors found that almost all the pathogenic mutations expressed to a significantly lower extent. Additionally, they utilized molecular dynamics simulation studies to show the mechanism of the peptide corresponding to the AH107-125 binding to lipid membranes. Overall, the manuscript is very well written, is thorough, and uncovers the function of the key AH107-125 region of a critical enzyme in the visual cycle. This reviewer would like to thank the authors for the final figure, as the cartoon model of the mechanism of membrane binding is clear and concise and it fits nicely with the strong data shown throughout the manuscript.

Minor changes to consider in a revision include:

1. In the abstract, 6th line should have an "alpha" instead of an a.
2. In Figure 1, it would be helpful for the reader to include the highest resolution of the crystal structure of RPE65 and highlight the AH107-125 region to better discern the relationship between the AH and the holoprotein.
3. In Figure 2, it would be helpful to have the font sizes bigger in the CD data and in the chart (Figure 2D).
4. Likewise, in Figure 3, bigger font sizes are needed, especially in B and in the graph for C.

Reviewer #2 (Comments to the Authors (Required)):

SUMMARY. This manuscript addresses the important and unresolved question of how RPE65 interacts with membranes. RPE65 is a well-studied peripheral membrane protein which encodes an essential enzymatic activity required for vertebrate vision. This manuscript's strength is based on use of multiple independent approaches. This is also its main weakness, because not all approaches have been implemented with equal rigor. By addressing some of the softer aspects, the authors would significantly improve consistency and overall quality. The work is clearly suited to publication in LSA, as it provides sufficient evidence to demonstrate that RPE 65 amino acids107-125 encode an inducible AH membrane interaction domain that is important for protein function, and that palmitoylation of a key cysteine (112) within this region promotes a stable secondary structure. This advance provides a plausible model for RPE65 membrane association and could facilitate new studies that could elucidate the mechanisms underlying substrate entry and product release.

CONCERNS TO BE ADDRESSED

- a. The lack of line numbers makes it harder than necessary to provide feedback; a revision should include these.
- b. Spaces are needed in front of (first parenthesis) citations.
- c. Introduction, p2, paragraph 4: "Paradox" appears to be misused; the authors instead seem to be referencing a "knowledge gap".
- d. Figure 1: some font sizes are too small and impair legibility
- e. Results, p3, paragraph 2: "The CD spectrum of AH107-125 peptide showed that the peptide is poorly soluble..." It's unclear how CD data was used to draw this conclusion; the method used should be described. This is an important point, because accurate protein quantification is required for obtaining high-quality CD data. The most rigorous quantification method for this

purpose is amino acid analysis, though a published experimentally determined extinction coefficient is also acceptable. Finally, protein secondary structure can be protein concentration dependent; were any concentrations other than 100 μ M peptide analyzed?

f. Results, p4, paragraph 4: This paragraph is unfocused and hard to follow; the logic flow needs to be improved.

g. Results, p4, paragraph 5: This paragraph belongs earlier in the Results section; I suggest moving it in front of the current paragraph 4. Concentration of LUVs should be given.

h. Results, p4, paragraph 6: Given that the approach produces three fractions, and that assaying all fractions gives an indication of the robustness of the membrane association, why is data only provided for the top fractions? Likewise, the % LUVs found in the top (vs other) fractions should be given. Regarding the corresponding Fig 3 - I don't see the value of including the ugly anti-HIS blot; I suggest it be removed and only the anti-MBP blot be retained. Presumably the bar graph quantitation was derived from those data? The figure legend should be edited to clarify that the lanes contain technical sample replicates.

i. Results, p5, paragraph 8: The level of rigor applied for the IHC analyses falls below that used for other approaches in the manuscript, and below what is current standard practice. This type of study is particularly prone to sampling bias, and to non-linear relationships between fluorescence intensity and protein concentration. Stating that "slides were analyzed by confocal laser scanning microscopy" is not adequate. Methods for avoiding sampling bias and non-linear relationships between fluorescence intensity and protein concentration should be applied (and described in Methods and Materials, paragraph 16), and some form of quantitative analysis applied. If tightened up, this type of analysis could also be productively applied to some of the pathogenic RPE mutations in the future.

j. Results, p6, paragraph 14: I'm not convinced that the MD simulation provides any weight to the conclusions derived for the WT peptide, from the wet experimental data. Do the in silico simulation results offer a new prediction that can be tested via wet lab methods? Conversely, do the wet lab findings make a strong prediction that can be examined in silico? IE - that palmitoylation should increase the membrane partition coefficient?

k. Discussion, p6, paragraph 1: "Interaction with RPE sER membranes" typo

l. Discussion, p8, paragraph 6: The finding that palmitoylation alone induces local folding is perhaps the key new finding reported. Its role is suggested to be "insertion of the AH along with the palmitoylation of C112 residue is the critical step in the proper orientation of substrate-binding cleft onto the membrane." Since there's no evidence offered that AH orientation in the membrane-bound state is affected by palmitoylation, perhaps this is best stated as a hypothesis. In the same paragraph, it might be interesting to offer a speculation as what function RPE65 desorption from the membrane may have.

m. Materials and Methods, p9, paragraph 2: determination method for final DOPC LUV concentration should be given. Materials and Methods, p9, paragraph 3: DOPC LUV concentration(s) used for CD should be given.

n. Materials and Methods, p9-11, paragraphs 3, 4, 9, 12, 14: sources (or references for custom-made Abs) should be provided for: DOPC, DDM, CHAPS, anti-HIS, anti-MBP, anti-RPE65, and anti-calreticulin antibodies.

Reviewer #3 (Comments to the Authors (Required)):

The study by Dr. Redmond and his team analyzes the role of a putative amphipathic alpha helix (AH) that may convey membrane binding of RPE65. RPE65 is the retinoid isomerase in the visual cycle and catalyzes the conversion of retinyl palmitate into 11-cis-retinol and palmitate. The authors conduct an elegant series of experiments to demonstrate that palmitoylation of this amino acid sequence is critical for AH formation and membrane binding. Furthermore, the authors show that the AH is sufficient to target a given protein to the ER, indicating that the AH acts as membrane targeting sequence. Disruption of the AH sequence by site directed mutagenesis alters membrane association and enzymatic activity of RPE65. Finally, molecular modelling indicates that the AH inserts into a membrane double layer, thereby corroborating the outcome of their biochemical and cell based assays.

Overall, the study is timely and well conducted. The outcomes of the experiments are convincing and sound. The demonstration that the AH is sufficient and necessary for membrane association of RPE65 is of importance for the visual cycle field and will stimulate further research to clarify important regulatory aspects of the visual cycle. This reviewer raises a few concerns and asks for some modifications of the figures to improve this already excellent study.

Suggestions and concerns:

Abstract: The first sentence of the abstract needs rewording. Use 'retinoid cycle' or 'visual cycle' but not both together. Additionally, the retinoid cycle takes place between retina and RPE.

Abstract: Usually, in silico findings are validated by biochemical experiment and not vice versa. Consider to rephrase the

sentence in the abstract.

Introduction: The reviewer suggests to use 'CCD' and not 'CO' for consistency with the literature. There is no need to introduce an additional term for the carotenoid cleavage dioxygenase (CCD) enzyme family to which RPE65 belongs.

Introduction, last paragraph: Which paradox is meant here? The topic of the manuscript is rather an unresolved question than a paradox.

Additionally, the abstract and other parts of the manuscript state that the AH motif regulates the visual cycle. The reviewer agrees that membrane binding of RPE65 is critical for its enzymatic activity, however, what is the evidence of a regulatory role of the AH? Previously, it was proposed that a light-dependent palmitoylation triggers membrane binding of RPE65 and regulates the visual cycle. However, this proposal has been disproven (see also discussion part of the present manuscript). The reviewer suggests to either provide experimental evidence for the proposed regulatory role of the AH or to omit the statement from the manuscript.

Result part, Pages 3 to 4: It is shown that a synthetic AH peptide binds to artificial membranes. Furthermore, it is shown that palmitoylation is critical for this binding. The reviewer wonders whether these conditions are sufficient for RPE65 enzymatic activity. In other words is recombinant and/or native RPE65 enzymatically active when incubated in the presence of SDS or DDM micelles? This would clearly strengthen the conclusion of this experiment.

Page 5: Introduce abbreviations such as HAM when they first appear in the text.

Discussion: Can the authors speculate how the AH targets RPE65 specifically to the ER. How is this ER specificity determined? Does it also target RPE65 to other membranes? Naively asked, is this targeting dependent on the substrate as indicated in the summary figure.

Regarding reference Xue et al. (2004): This reviewer is wondering why the authors discuss the outcome of a study that has been disproven and seemingly was based on manufactured data? The regulatory role of the AH needs to be demonstrated by accurate experiments.

Figure 1, panel E: Increase font size.

Figure 2: Panel D is difficult to read and would benefit from increased font size. Panel E: The insets are not readable and need to be enlarged.

Figure 4: The caption of the size bars are not readable in panel A. Additionally, the experiment with RPE65 in panel B is not very convincing. Is there a better picture that shows ER localization of RPE65? What is the laminar staining at the bottom of the cell?

Figure 5: Why does the C112A mutant show about 20% palmitoylation? Are there other palmitoylation sites in RPE65?

Additionally, many pathological RPE65 mutations display reduced membrane binding and enzymatic activity. These mutations are not necessarily localized in the AH. For instance a D477G mutation does affect both characteristics. It might be worthwhile to at least discuss this issue. Are there other mechanisms that disrupt membrane binding of RPE65? This point needs to be addressed.

Reviewer #1 (Comments to the Authors (Required)):

RPE65 retinol isomerase is essential in the visual cycle of the retinal pigmented epithelium (RPE) in vertebrate retina. Membrane association is critical, although how it is associated to the membrane remains unclear. The authors of this work analyzed the amino acid residues aa107-125 of RPE65, an inherently disordered region of the protein in the crystal structure, to uncover the role of this region in the overall function of the protein. Through elegant and classical biochemical and biophysical techniques, they have found that the aa107-125 can act as a "membrane sensor" that can switch between an amphipathic helix (AH) and a disordered loop. The authors assessed membrane association of the AH region using alanine-scanning mutagenesis of the protein in the AH region, and found that for four mutants (F108A, D110A, P111A, and R118A) they observed a loss of RPE65 protein in the membrane fraction through subcellular fractionation assays, as well as a severe loss in RPE65 function in the presence of those mutants. The circular dichroism (CD) data of the AH with and without the palmitoylation at C112 are compelling and show the propensity of the palmitoylated peptide to display a significantly ordered AH with palmitoylation. Through confocal laser scanning microscopy of transfected tissue culture cells expressing GFP fusions of the AH, the authors found that almost all the pathogenic mutations expressed to a significantly lower extent. Additionally, they utilized molecular dynamics simulation studies to show the mechanism of the peptide corresponding to the AH107-125 binding to lipid membranes. Overall, the manuscript is very well written, is thorough, and uncovers the function of the key AH107-125 region of a critical enzyme in the visual cycle. This reviewer would like to thank the authors for the final figure, as the cartoon model of the mechanism of membrane binding is clear and concise and it fits nicely with the strong data shown throughout the manuscript.

Response: We thank the reviewer for the positive comments made.

Minor changes to consider in a revision include:

1. In the abstract, 6th line should have an "alpha" instead of an a.

Response: Done as requested.

2. In Figure 1, it would be helpful for the reader to include the highest resolution of the crystal structure of RPE65 and highlight the AH107-125 region to better discern the relationship between the AH and the holoprotein.

Response: Thank you for this comment. In the revised Figure 1 we have now included the three-dimensional structure of bovine RPE65 and the crystallographically unresolved region, amino acid 107-125, highlighted in the red color is generated using secondary structure prediction server, I-TASSER.

3. In Figure 2, it would be helpful to have the font sizes bigger in the CD data and in the chart (Figure 2D).

Response: We have now made the font sizes bigger in the revised Figure 2.

4. Likewise, in Figure 3, bigger font sizes are needed, especially in B and in the graph for C.

Response: We have now made the font sizes bigger in the revised Figure 3.

Reviewer #2 (Comments to the Authors (Required)):

SUMMARY. This manuscript addresses the important and unresolved question of how RPE65 interacts with membranes. RPE65 is a well-studied peripheral membrane protein which encodes an essential enzymatic activity required for vertebrate vision. This manuscript's strength is based on use of multiple independent approaches. This is also its main weakness, because not all approaches have been implemented with equal rigor. By addressing some of the softer aspects, the authors would significantly improve consistency and overall quality. The work is clearly suited to publication in *LSA*, as it provides sufficient evidence to demonstrate that RPE65 amino acids 107-125 encode an inducible AH membrane interaction domain that is important for protein function, and that palmitoylation of a key cysteine (112) within this region promotes a stable secondary structure. This advance provides a plausible model for RPE65 membrane association and could facilitate new studies that could elucidate the mechanisms underlying substrate entry and product release.

Response: We thank the reviewer for the positive comments made.

CONCERNS TO BE ADDRESSED

a. The lack of line numbers makes it harder than necessary to provide feedback; a revision should include these.

Response: Done as requested.

b. Spaces are needed in front of (first parenthesis) citations.

Response: Done as requested.

c. Introduction, p2, paragraph 4: "Paradox" appears to be misused; the authors instead seem to be referencing a "knowledge gap".

Response: Done as suggested.

d. Figure 1: some font sizes are too small and impair legibility

Response: Adjustments made as requested.

e. Results, p3, paragraph 2: "The CD spectrum of AH107-125 peptide showed that the peptide is poorly soluble..." It's unclear how CD data was used to draw this conclusion; the method used should be described.

Response: This was an unfortunate misstatement that conveyed the wrong intent. The peptide is soluble in aqueous buffer/water, it just lacked appreciable secondary structure.

This is an important point, because accurate protein quantification is required for obtaining high-quality CD data. The most rigorous quantification method for this purpose is amino acid analysis, though a published experimentally determined extinction coefficient is also acceptable. Finally, protein secondary structure can be protein concentration dependent; were any concentrations other than 100 μ M peptide analyzed?

Response: For accurate peptide quantification, the extinction coefficient of the peptide was used as calculated by the ExPASy ProtParam (<https://web.expasy.org/protparam/>) tool. The extinction coefficient of the peptide corresponding to the crystallographically unresolved region (107-125 aa) of RPE65 protein is $1490 \text{ M}^{-1}\text{cm}^{-1}$ at 280 nm. We have now included this in the Materials and Methods section of the revised manuscript. We had tested 25 μM and 50 μM concentration of peptides and observed very high HT voltage below 220 nm which resulted in high noise-to-signal ratio (Fig. R1) whereas at 100 μM , the HT voltage was in the range, so we used this concentration for further analysis.

Fig. R1 Far-UV-CD spectra of non-palmitoylated and palmitoylated peptide corresponding to the crystallographically unresolved region (107-125 aa) of RPE65 protein.

f. Results, p4, paragraph 4: This paragraph is unfocused and hard to follow; the logic flow needs to be improved.

Response: To address these concerns we have significantly modified the text and hope that the revision meets with the reviewer's approval:

'Gel filtration chromatography was used to further investigate the interaction of the RPE65¹⁰⁷⁻¹²⁵ peptide with detergent. We found that non-palmitoylated RPE65¹⁰⁷⁻¹²⁵ peptide dissolved in DDM micelles elutes at ~17.9 min, which corresponds to a molecular weight of ~49 kDa that is roughly equal to the molecular weight of detergent micelles. In contrast, in the absence of detergent, non-palmitoylated RPE65¹⁰⁷⁻¹²⁵ peptide elutes at ~30 min which is much later than expected, considering its low molecular weight (2300 Da; Fig. 2 E, left panel). This suggests that the non-palmitoylated peptide interacts with the chromatographic media, probably via hydrophobic interactions. Indeed, detergent was necessary to elute completely all the non-specifically bound non-palmitoylated peptide from the gel filtration column. When CD spectra of the eluted peptide were recorded, a significant change from unstructured to α -helical structure of the non-palmitoylated peptide was detected in the presence of DDM detergent micelles (Fig. 2 E, insets in left panels). In contrast, the gel filtration profiles of C112-palmitoylated RPE65¹⁰⁷⁻¹²⁵ peptide were quite different. In the presence of DDM, the C112-palmitoylated peptide elutes earlier (~17.6 min) compared to the peptide in the absence of detergent (~23.1 min; Fig. 2 E, right panels and insets). However,

the latter elution time is much earlier than that of the non-palmitoylated peptide in the absence of detergent (Fig. 2 E, left panel). These data indicate further stabilization of the α -helical structure of the palmitoylated peptide upon binding to the detergent micelles. The calculated molecular weights of the detergent micelle-bound non-palmitoylated and palmitoylated RPE65¹⁰⁷⁻¹²⁵ peptide from the calibration curve is ~50 kDa (Fig. S1). These data are concordant with a strong propensity of this peptide to bind lipid-mimicking detergents.'

In addition, we made an addition to the legend for revised Figure 2 E to clarify the CD spectra insets.

g. Results, p4, paragraph 5: This paragraph belongs earlier in the Results section; I suggest moving it in front of the current paragraph 4. Concentration of LUVs should be given.

Response: This has been done as suggested. The stock concentration of LUVs used was 2 mM which is now included in the Materials and Methods section of the revised manuscript.

h. Results, p4, paragraph 6: Given that the approach produces three fractions, and that assaying all fractions gives an indication of the robustness of the membrane association, why is data only provided for the top fractions? Likewise, the % LUVs found in the top (vs other) fractions should be given. Regarding the corresponding Fig 3 - I don't see the value of including the ugly anti-HIS blot; I suggest it be removed and only the anti-MBP blot be retained. Presumably the bar graph quantitation was derived from those data? The figure legend should be edited to clarify that the lanes contain technical sample replicates.

Response: We followed the liposome flotation method as described in www.bio-protocol.org/e2169 in which they assessed only the top fraction for liposome-bound protein. We repeated the experiment, assessed all the three fractions by SDS-PAGE and Coomassie staining. We estimated the % LUV in the top and other fractions using EnzyChrom™ phospholipid assay kit. We have now included the results in the revised draft. In addition to the requested corrections, we took the opportunity to re-write the paragraph in a clearer way. In conjunction of the changes in the paragraph we reversed the order of panels A and B of revised Fig. 3. The paragraph now reads:

'To extend these studies we next performed a sucrose gradient liposome co-flotation assay. In this assay, membrane-binding proteins float during centrifugation with liposomes in the upper, lower-density fraction, while soluble proteins remain in the lower, higher-density fraction (Fig. 3 A). A fusion construct of 6xHis-maltose-binding protein (MBP) and RPE65¹⁰⁷⁻¹²⁵ AH peptide (6xHis-MBP-AH¹⁰⁷⁻¹²⁵) was generated and over-produced heterologously in Escherichia coli. The 6xHis-MBP-AH¹⁰⁷⁻¹²⁵ protein was purified to homogeneity with an apparent molecular weight of ~46 kDa on SDS-PAGE gel, slightly higher than the purified 6xHis-MBP protein (Fig. 3 B). For the assay, 6xHis-MBP and 6xHis-MBP-AH¹⁰⁷⁻¹²⁵ purified proteins were mixed, separately, with fluorescent 1-Oleoyl-2-[12-[(7-nitro-2-1,3-benzoxadiazol-4-yl)amino]dodecanoyl]-sn-Glycero-3-Phosphocholine (NBD:PC-DOPC) liposomes and resolved in sucrose density gradients. After centrifugation, the upper fractions containing the liposomes and the middle and bottom fractions were collected and subjected to SDS-PAGE and Coomassie staining. As shown in Fig. 3 C and D, 6xHis-MBP-AH¹⁰⁷⁻¹²⁵ protein showed an increase level of protein (~5-fold) compared to the low level of control 6xHis-MBP protein in the liposome-containing upper fraction, indicating the binding of RPE65¹⁰⁷⁻¹²⁵ peptide to the liposomal membranes. Furthermore, we assessed the presence of liposomes in each fraction and observed almost ~100% liposomes were fractionated in the top fraction upon centrifugation (Fig 3 E).'

i. Results, p5, paragraph 8: The level of rigor applied for the IHC analyses falls below that used for other approaches in the manuscript, and below what is current standard practice. This type of study is particularly prone to sampling bias, and to non-linear relationships between fluorescence intensity and protein concentration. Stating that "slides were analyzed by confocal laser scanning microscopy" is not adequate. Methods for avoiding sampling bias and non-linear relationships between fluorescence intensity and protein

concentration should be applied (and described in Methods and Materials, paragraph 16), and some form of quantitative analysis applied. If tightened up, this type of analysis could also be productively applied to some of the pathogenic RPE mutations in the future.

Response: We apologize for not doing a quantitative analysis of the IHC data from the outset. As suggested, we have now performed the quantitative analysis of the GFP fluorescence intensity in the nucleus and the ER compartment of cells (~20 cells) expressing GFP and GFP-AH¹⁰⁷⁻¹²⁵ constructs using ImageJ (NIH) software. The box and whisker graph is plotted showing the percentage background corrected fluorescence intensity (Fig. 4C in the revised draft) and the method is also included in the Materials and Methods section. Thank you for this suggestion, we are interested in understanding the mechanism of AH-mediated targeting of RPE65. One of our future plans is to use the same approach to study the pathogenic RPE65 amphipathic helix (AH) mutations using IHC methods and to analyze the role of AH on the RPE65 targeting to the ER.

j. Results, p6, paragraph 14: I'm not convinced that the MD simulation provides any weight to the conclusions derived for the WT peptide, from the wet experimental data. Do the in-silico simulation results offer a new prediction that can be tested via wet lab methods? Conversely, do the wet lab findings make a strong prediction that can be examined in silico? IE - that palmitoylation should increase the membrane partition coefficient?

Response: While we agree in general terms with the reviewer's comment, we think that the MD simulation results do provide a better insight into the mechanism of AH of RPE65-membrane binding. From the "wet" experimental results, we conclude that the amphipathic helix interacts with the lipid bilayer, however, the MD simulation data hints at the individual steps and transitions involved in the AH-RPE65 binding to the lipid bilayer. As suggested, we hope to discern the effect of palmitoylation on the membrane partition coefficient for the AH peptide/protein. We assume that AH insertion into the lipid bilayer allows proper orientation of the hydrophobic tunnel of RPE65 to extract its substrate, atRE, which are lipophilic in nature and stored in lipid bilayer. Keeping this in mind, our future experimental aims will include performing MD simulation of the complete RPE65 protein with the lipid bilayer containing atRE. We would also study the effect of AH mutations on binding. We will also attempt fluorescence quenching experiments with both non-palmitoylated and palmitoylated forms of fluorescently labeled AH peptides to understand the mechanism of RPE65-membrane binding in a more detailed manner.

k. Discussion, p6, paragraph 1: "Interaction with RPE sER membrane" typo.

Response: We have corrected the typo to "Interaction with RPE smooth ER membranes"

l. Discussion, p8, paragraph 6: The finding that palmitoylation alone induces local folding is perhaps the key new finding reported. Its role is suggested to be "insertion of the AH along with the palmitoylation of C112 residue is the critical step in the proper orientation of substrate-binding cleft onto the membrane." Since there's no evidence offered that AH orientation in the membrane-bound state is affected by palmitoylation, perhaps this is best stated as a hypothesis. In the same paragraph, it might be interesting to offer a speculation as what function RPE65 desorption from the membrane may have.

Response: As suggested, we have formulated this statement as a hypothesis:

"We hypothesize that this insertion of the AH along with the palmitoylation of C112 residue is the critical step in the proper orientation of substrate-binding cleft onto the membrane allowing RPE65 to extract its highly lipophilic substrate, atRP."

Also, as prompted, we offer a speculation as to the role/process of RPE65 desorption. We know from our previous work (Uppal et al, 2019) that the level of RPE65 palmitoylation is reduced in the presence of active LRAT. In that paper we speculated that the palmitoyl group could be donated to/removed by LRAT to esterify at-retinol. If that were to happen, we can speculate here that the AH would be weakened and, therefore, RPE65 could desorb. This could be required for transfer of 11-cis retinol to CRALBP. The mechanism of 11-cis retinol transfer from RPE65 is currently not known for certain: whether it is passed directly to RDH5 in the membrane phase for oxidation or if it is passed to CRALBP in the soluble phase,

which then presents it to RDH5 for oxidation. The latter scenario is more likely as presence of CRALBP is required for progression of isomerization. Absence of CRALBP stalls isomerization. The following text has been added:

'How does this proceed and what is the functional role of desorption? We know from our previous work (Uppal et al, 2019) that RPE65 palmitoylation is reduced in the presence of active LRAT. In that paper we speculated that the palmitoyl group could be donated to/removed by LRAT to esterify all-trans retinol. Alternatively, APT deacylation may occur. Either case would weaken AH and, therefore, RPE65 could disassociate from the membrane. We speculate that this desorption could be required for transfer of 11-cis retinol from the binding tunnel of RPE65 to 11-cis-specific cellular retinal-binding protein (CRALBP; RLBPI). CRALBP is an obligate component of the visual cycle, playing a key role in the downstream processing of 11-cis retinol (Saari et al., 1994). The 11-cis retinol is thought to exit the active site first (Kiser et al, 2015). How 11-cis retinol transfers from RPE65 is currently unclear: whether it is passed directly for oxidation to RDH5 in the membrane phase or if it is passed to CRALBP in the soluble phase, which then presents it to RDH5 for oxidation. The latter scenario is more likely as absence of CRALBP stalls progress of isomerization (Saari et al., 2001). Upon transfer of its 11-cis retinol product to CRALBP, RPE65 would be ready for another isomerization cycle after the removal of the palmitate product from the cleft by some as yet unknown mechanism. These possibilities require clarification.'

m. Materials and Methods, p9, paragraph 2: determination method for final DOPC LUV concentration should be given. Materials and Methods, p9, paragraph 3: DOPC LUV concentration(s) used for CD should be given.

Response: We have now included in the revised draft, in Materials and Methods, the phospholipid determination method for measuring the DOPC LUV concentration used in the CD and liposome co-floitation experiments.

n. Materials and Methods, p9-11, paragraphs 3, 4, 9, 12, 14: sources (or references for custom-made Abs) should be provided for: DOPC, DDM, CHAPS, anti-HIS, anti-MBP, anti-RPE65, and anti-calreticulin antibodies.

Response: Completed as requested.

Reviewer #3 (Comments to the Authors (Required)):

The study by Dr. Redmond and his team analyzes the role of a putative amphipathic alpha helix (AH) that may convey membrane binding of RPE65. RPE65 is the retinoid isomerase in the visual cycle and catalyzes the conversion of retinyl palmitate into 11-cis-retinol and palmitate. The authors conduct an elegant series of experiments to demonstrate that palmitoylation of this amino acid sequence is critical for AH formation and membrane binding. Furthermore, the authors show that the AH is sufficient to target a given protein to the ER, indicating that the AH acts as membrane targeting sequence. Disruption of the AH sequence by site directed mutagenesis alters membrane association and enzymatic activity of RPE65. Finally, molecular modelling indicates that the AH inserts into a membrane double layer, thereby corroborating the outcome of their biochemical and cell-based assays.

Overall, the study is timely and well conducted. The outcomes of the experiments are convincing and sound. The demonstration that the AH is sufficient and necessary for membrane association of RPE65 is of importance for the visual cycle field and will stimulate further research to clarify important regulatory aspects of the visual cycle. This reviewer raises a few concerns and asks for some modifications of the figures to improve this already excellent study.

We thank the reviewer for the positive comments made.

Suggestions and concerns:

Abstract: The first sentence of the abstract needs rewording. Use 'retinoid cycle' or 'visual cycle' but not both together. Additionally, the retinoid cycle takes place between retina and RPE.

Response: Changed as requested.

Abstract: Usually, in silico findings are validated by biochemical experiment and not vice versa. Consider to rephrase the sentence in the abstract.

Response: Thank you for the valuable comment. We changed “strongly validating” to “supporting”.

Introduction: The reviewer suggests to use 'CCD' and not 'CO' for consistency with the literature. There is no need to introduce an additional term for the carotenoid cleavage dioxygenase (CCD) enzyme family to which RPE65 belongs.

Response: As suggested, we changed CO to CCD.

Introduction, last paragraph: Which paradox is meant here? The topic of the manuscript is rather an unresolved question than a paradox.

Response: As suggested by this Reviewer and Reviewer #1, we changed “paradox” to “knowledge gap”.

Additionally, the abstract and other parts of the manuscript state that the AH motif regulates the visual cycle. The reviewer agrees that membrane binding of RPE65 is critical for its enzymatic activity, however, what is the evidence of a regulatory role of the AH?

Previously, it was proposed that a light-dependent palmitoylation triggers membrane binding of RPE65 and regulates the visual cycle. However, this proposal has been disproven (see also discussion part of the present manuscript). The reviewer suggests to either provide experimental evidence for the proposed regulatory role of the AH or to omit the statement from the manuscript.

Response: We do not go so far as to claim that the AH, per se, plays a regulatory role in the visual cycle. What we are saying is that the AH plays an important role in the function of RPE65 by targeting it to the membrane where its substrate is located. Since RPE65 is the rate-limiting step in the visual cycle, presence of the AH is important for the function of RPE65 in the visual cycle, but by itself the AH does not regulate the visual cycle. To clarify this and mitigate any potential misunderstandings we have rewritten the last sentence of the Introduction to read:

‘Overall, our work establishes that the AH¹⁰⁷⁻¹²⁵ region in RPE65 protein is a key structural element of this protein that serves as an intrinsic membrane-targeting motif and, thus, is essential to the function of RPE65.’

In addition, the first sentence of the legend to Fig. 7 has been changed to:

‘We propose that the aa107-125 sequence is the key structural element of RPE65 that serves as an intrinsic membrane-targeting motif thereby regulating the function of RPE65 in the visual cycle.’

On the other hand, the statement at issue was not included in the Abstract. We address the issue of Xue et al.’s “light-dependent palmitoylation” later in our responses.

Result part, Pages 3 to 4: It is shown that a synthetic AH peptide binds to artificial membranes. Furthermore, it is shown that palmitoylation is critical for this binding. The reviewer wonders whether these conditions are sufficient for RPE65 enzymatic activity. In other words is recombinant and/or native RPE65 enzymatically active when incubated in the presence of SDS or DDM micelles? This would clearly strengthen the conclusion of this experiment.

Response: We do not perform RPE65 isomerization activity experiments in the presence of SDS. We used SDS micelles as a membrane mimetic to monitor the changes in the secondary structure of synthetic AH-RPE65¹⁰⁷⁻¹²⁵ peptide. We also used DDM micelles and DOPC liposomes and these reagents are included in an assay that we use in the laboratory (e.g., used in our paper at PMID: 26719343) to assess RPE65 enzymatic activity, and RPE65 is functionally active in these conditions. However, this particular assay was not employed in this study. Briefly, in this RPE65 enzymatic assay we use DDM micelles to solubilize the recombinant or native RPE65 in the cell membrane fractions, which are then mixed with DOPC liposomes incorporating RPE65’s substrate, all-*trans* retinyl palmitate.

Page 5: Introduce abbreviations such as HAM when the first appear in the text.

Response: Done as requested.

Discussion: Can the authors speculate how the AH targets RPE65 specifically to the ER. How is this ER specificity determined? Does it also target RPE65 to other membranes?

Response: These are all interesting questions that deserve much more detailed analysis and further research specifically in RPE. The answers may relate to the kind of lipid membranes occurring in the RPE ER. In general terms, ER membranes are part of a lipid “territory” comprising ER-nuclear envelope-cis-Golgi, while the other major “territory” centers on trans-Golgi-plasma membrane- endosomes. The difference between the two relates to lipid packing— the former being characterized by membranes with little anionic charge and loose packing of ‘conical’ and ‘cylindrical’ phospholipids (i.e., with “packing defects”) while the latter have high levels of ‘cylindrical’ phospholipids, sterols, and sphingolipids, and so are quite tightly packed. It is known that AHs favor binding to membranes with packing defects, i.e., the ER-nuclear envelope-cis-Golgi territory. It is also known that different kinds of AH localize to different regions of the territory, depending both on the relative charge of the polar face residues and on the relative size and hydrophobicity of the hydrophobic face residues. The precise lipid analysis of RPE sER (microsomal membranes) is not known, but we plan to determine this. Does the particular conformation of the RPE65 AH¹⁰⁷⁻¹²⁵ target it to a particular region of the RPE ER? We don’t have the answer to this question, yet. Another complication will be admixture of retinyl palmitate. We plan to study all these aspects. However, we can say that the AH¹⁰⁷⁻¹²⁵ targets to ER rather than to the PM territory. To address these points we have added to the Discussion:

‘The universe of the AH repertoire is wide– it is known that some AHs favor binding to membranes that are curved (Drin et al., 2007; Bigay et al, 2005), have packing defects and/or increased surface charge

(Cornell, 2016), or to lipid droplets (Pataki et al 2018; Olarte et al, 2022), among others. Also, a cholesterol sensing AH in squalene monooxygenase has been identified that converts into a degranon when ER membrane cholesterol becomes high (Chua et al, 2017). ER membranes are a part of a cellular lipid “territory” comprising ER-nuclear envelope (NE)-cis-Golgi, while the other major “territory” centers on trans-Golgi-plasma membrane (PM)- endosomes (Bigay and Antonny, 2012; Jackson et al, 2016). The difference between the two relates to lipid packing– the former being characterized by membranes with little anionic charge and loose packing of ‘conical’ and ‘cylindrical’ phospholipids and low amounts of sterols and sphingolipids (i.e., with “packing defects”), while the latter have high levels of ‘cylindrical’ phospholipids, sterols, and sphingolipids, and so are quite tightly packed (Jackson et al, 2016). It is also known that different kinds of AH localize to different membrane regions, depending both on the relative charge of the polar face residues and on the relative size and hydrophobicity of the hydrophobic face residues (Bigay and Antonny, 2012; Pranke et al, 2011). Does the particular conformation of the RPE65 AH¹⁰⁷⁻¹²⁵ target it to a particular region of the RPE ER? We don’t have the answer to this question, but experiments to address this are underway. However, we can conclude that the AH¹⁰⁷⁻¹²⁵ sequence targets to the ER rather than to the PM territory.’

Naively asked, is this targeting dependent on the substrate as indicated in the summary figure.

Response: This is something that we do not know yet but, given the points addressed above, is not outside the bounds of possibility. We are addressing this in ongoing experiments.

Regarding reference Xue et al. (2004): This reviewer is wondering why the authors discuss the outcome of a study that has been disproven and seemingly was based on manufactured data?

Response: We include this reference for the sake of historical accuracy/completeness. In light of this comment, we have modified the text to put it in context:

‘While Xue et al. proposed a “palmitoylation switch” mechanism to account for RPE65’s dual nature of soluble (“sRPE65”) and membrane-associated forms (“mRPE65”); Xue et al., 2004), it was on the basis of data later shown to be erroneous (Takahashi et al., 2009). Despite this false lead, palmitoylation (at C112) does play an important role in RPE65 membrane association and enzymatic function (Takahashi et al., 2009; Uppal et al., 2019a).

The regulatory role of the AH needs to be demonstrated by accurate experiments.

Response: We fully agree with the reviewer’s sentiment and we hope that our studies help to accomplish this.

Figure 1, panel E: Increase font size.

Response: Done as requested.

Figure 2: Panel D is difficult to read and would benefit from increased font size. Panel E: The insets are not readable and need to be enlarged.

Response: We have now made the required changes in the revised Figure 2.

Figure 4: The caption of the size bars are not readable in panel A. Additionally, the experiment with RPE65 in panel B is not very convincing. Is there a better pictures that shows ER localization of RPE65? What is the laminar staining at the bottom of the cell?

Response: We are sorry for this oversight. We have now increased the font size of the size bars and replaced the images as well in revised Figure 4, panel A. We fully agree with the reviewer that the experiment with RPE65 in panel B is not convincing. We would like to mention here that we always have trouble co-transfecting the COS7 cells with RPE65 along with DsRedER plasmid. We observed very few cells expressing the RPE65 protein, the expression level was also not good and furthermore, we used the custom-made polyclonal RPE65 antibody which we think was not optimal for RPE65 staining. The laminar staining at the bottom of the cells may be non-specific signal from the antibody.

To address this issue, we repeated the experiment where COS7 cells were singly-transfected with RPE65 plasmid which showed a considerable increase in the transfection efficiency and expression of RPE65 protein. For the primary RPE65 antibody, we used commercial rabbit monoclonal antibody (Abcam) which provides much better signal compared to previously used custom-made polyclonal antibody. We stained the ER compartment using a red fluorescence ER staining kit (Cytopainter; Abcam). We obtained better quality images and these are included in the revised draft

Fig R2. Validation of commercial RPE65 mouse monoclonal antibody. Western blot image shows the RPE65 protein band in the COS7-RPE65 transfected samples and not in the control untransfected sample. We also tested the antibody for immunostaining using dilutions 1:500 and 1:250 and observed green immunoreactive fluorescence signal only in the RPE65-transfected samples.

Figure 5: Why does the C112A mutant show about 20% palmitoylation? Are there other palmitoylation sites in RPE65?

Response: Thank you for this question; we are glad to clarify this point. We performed the quantitative analysis of palmitoylation level of AH mutants compared to the palmitoylation level of wild-type RPE65. For this, we kept the palmitoylation level of wild-type RPE65 to 100% but, in actual, the palmitoylation level of wild-type RPE65 is 25 % (calculated using ratio of protein band intensity in hydroxylamine treated sample to total input protein used for the acyl-RAC palmitoylation detection assay) indicating the dynamic

nature of RPE65 palmitoylation. We have previously shown (Uppal et al, Sci Rep 2019, 9:2518; cited herein) that C112A mutant showed almost no palmitoylation with ~ 3% palmitoylation level. This 20% palmitoylation of C112A mutants in comparison to wild type RPE65 is almost negligible. In addition to C112, we have previously shown (Uppal et al, Sci Rep 2019, 9:2518; cited herein) that C146 is capable of being palmitoylated in RPE65 protein.

Below are the raw data values of the triplicate samples of WT and C112 A mutants:

	% Palmitoylation level (compared to total Input)			% Palmitoylation level (compared to WT)		
	WT	23.43	24.16667	27.71845	100.0059	99.99999
C112	3.333333	3.918514	3.325123	14.22677	16.21454	11.99606

Additionally, many pathological RPE65 mutations display reduced membrane binding and enzymatic activity. These mutations are not necessarily localized in the AH. For instance, a D477G mutations does affect both characteristics. It might be worthwhile to at least discuss this issue. Are there other mechanisms that disrupt membrane binding of RPE65? This point needs to be addressed.

Response: We thank the reviewer for this interesting comment. A recent article published by Wu et al., 2022 *PNAS* (<https://www.pnas.org/doi/pdf/10.1073/pnas.2115202119>) reported that D477G mutation induces protein aggregation, reduced cellular RPE65 stability, and altered RPE65 subcellular distribution which contribute to other mechanisms which can disrupt the membrane binding of RPE65 protein. In general, RPE65 is quite sensitive to missense mutations that play out in many different ways. While we feel that these otherwise interesting findings are not particularly germane to the focus of our study on the AH, we have added the following text to Paragraph 1 of the revised Discussion (ll. 393-397):

“While the AH appears to be the primary means by which RPE65 as a PMP binds to RPE sER, other surface residues or patches of residues may play a role (Kiser et al, 2009). For example, RPE65 missense mutations distant from aa107-125 have the capacity to modify its folding/stability in ways that may disrupt its membrane binding (Wu et al., 2022).”

September 22, 2022

RE: Life Science Alliance Manuscript #LSA-2022-01546-TR

Dr. T. Michael Redmond
National Institutes of Health
National Eye Institute, Lab of Retinal Cell & Molecular Biology
6 Center Drive
Building 6, Room 117A
Bethesda, Maryland 20892

Dear Dr. Redmond,

Thank you for submitting your revised manuscript entitled "An Inducible Amphipathic α -Helix Mediates Subcellular Targeting and Membrane Binding of RPE65". We would be happy to publish your paper in Life Science Alliance pending final revisions necessary to meet our formatting guidelines.

- please use the [10 author names, et al.] format in your references (i.e. limit the author names to the first 10)
- Figure 5B: splices in bottom row, 2nd and 3rd sections. Please provide source data for this figure panel

A. FINAL FILES:

B. MANUSCRIPT ORGANIZATION AND FORMATTING:

Sincerely,

Reviewer #2 (Comments to the Authors (Required)):

This revision has addressed my previous concerns and represents an important and rigorous new contribution to the RPE65 literature. Well done.

Reviewer #3 (Comments to the Authors (Required)):

RPE65 is the retinoid isomerase of the visual cycle and catalyzes the conversion of retinyl palmitate into 11-cis-retinol and palmitate. The authors conduct an elegant series of experiments to demonstrate that palmitoylation of a cysteine residue of an amphipathic alpha helix (AH) is critical and sufficient for membrane binding of this rate limiting enzyme of the visual cycle. The outcomes of the experiments is believable and justifies the main conclusions. The study clarifies an important aspect of RPE65's biochemistry and provides important new knowledge to the visual cycle research field.

The authors address the concerns/suggestions of this reviewer satisfactorily and present an improved revised version of the manuscript. The reviewer compliments the authors on this study and has no further comments.

September 28, 2022

RE: Life Science Alliance Manuscript #LSA-2022-01546-TRR

Dr. T. Michael Redmond
National Institutes of Health
National Eye Institute, Lab of Retinal Cell & Molecular Biology
6 Center Drive
Building 6, Room 117A
Bethesda, Maryland 20892

Dear Dr. Redmond,

Thank you for submitting your Research Article entitled "An Inducible Amphipathic α -Helix Mediates Subcellular Targeting and Membrane Binding of RPE65". It is a pleasure to let you know that your manuscript is now accepted for publication in Life Science Alliance. Congratulations on this interesting work.

DISTRIBUTION OF MATERIALS:

Again, congratulations on a very nice paper. I hope you found the review process to be constructive and are pleased with how the manuscript was handled editorially. We look forward to future exciting submissions from your lab.

Sincerely,
